# Efficient online algorithms for fast-rate regret bounds under sparsity

**Pierre Gaillard**
INRIA, ENS, PSL Research University
Paris, France
pierre.gaillard@inria.fr

**Olivier Wintenberger**
Sorbonne Université, CNRS, LPSM
Paris, France
olivier.wintenberger@upmc.fr

## Abstract

We consider the problem of online convex optimization in two different settings: arbitrary and i.i.d. sequence of convex loss functions. In both settings, we provide efficient algorithms whose cumulative excess risks are controlled with fast-rate sparse bounds. First, the excess risks bounds depend on the sparsity of the objective rather than on the dimension of the parameters space. Second, their rates are faster than the slow-rate $1/\sqrt{T}$ under additional convexity assumptions on the loss functions. In the adversarial setting, we develop an algorithm BOA+ whose cumulative excess risks is controlled by several bounds with different trade-offs between sparsity and rate for strongly convex loss functions. In the i.i.d. setting under the Łojasiewicz's assumption, we establish new risk bounds that are sparse with a rate adaptive to the convexity of the risk (ranging from a rate $1/\sqrt{T}$ for general convex risk to $1/T$ for strongly convex risk). These results generalize previous works on sparse online learning under weak assumptions on the risk.

## 1 Introduction

We consider the following setting of online convex optimization where a sequence of random convex loss functions $(\ell_t : \mathbb{R}^d \to \mathbb{R})_{t \geqslant 1}$ is sequentially observed. At each iteration $t \geqslant 1$, a learner chooses a point $\widehat{\theta}_{t-1} \in \mathbb{R}^d$ based on past observations $\mathcal{F}_{t-1} = \sigma\big(\{\ell_1, \ldots, \ell_{t-1}\}\big)$. The learner aims at minimizing the average excess risk defined as $\widehat{L}_T := (1/T) \sum_{t=1}^{T} \mathbb{E}_{t-1}\big[\ell_t(\widehat{\theta}_{t-1})\big]$ where $\mathbb{E}_{t-1} = \mathbb{E}[\,\cdot\,|\mathcal{F}_{t-1}]$. For any parameter $\theta$ in some reference set $\Theta \subset \mathbb{R}^d$, the average excess risk can be decomposed as the sum of the approximation-estimation errors:

$$\widehat{L}_T = \underbrace{\frac{1}{T} \sum_{t=1}^{T} \mathbb{E}_{t-1}\big[\ell_t(\theta)\big]}_{\text{approximation error}} + \underbrace{\frac{1}{T} \sum_{t=1}^{T} \mathbb{E}_{t-1}\big[\ell_t(\widehat{\theta}_{t-1})\big] - \frac{1}{T} \sum_{t=1}^{T} \mathbb{E}_{t-1}\big[\ell_t(\theta)\big]}_{\substack{\text{estimation error} \\ := R_T(\theta)}} . \tag{1}$$

Though the final goal is to minimize $\widehat{L}_T$, a common proxy is to upper-bound the estimation term $R_T(\theta)$ (also refereed to as average excess risk[1]) simultaneously for all $\theta \in \Theta$. If the loss functions are exp-concave and $\Theta$ is bounded, several sequential algorithms achieve the uniform bound[2] on the estimation term $R_T := \sup_{\theta \in \Theta} R_T(\theta) \leqslant \mathcal{O}(d/T)$; see [13]. In this paper, we are interested with non-uniform bounds on $R_T(\theta)$ increasing with the complexity of $\theta$. Such non-uniform bounds are called oracle inequalities and state that the learner achieves the best approximation-estimation

trade-off of (1). Using the $\ell_0$-norm to measure the complexity of $\theta$, we are looking for fast-rate sparse bounds of the form

$$R_T(\theta) \leqslant \tilde{\mathcal{O}}\left(\left(\frac{\|\theta\|_0}{T}\right)^{\frac{1}{2-\beta}}\right), \qquad \text{for any } \theta \in \Theta.$$

The parameter $\beta \in [0,1]$ depends on the convexity properties of the loss functions and will be specified later. We call fast-rate bound any bound which provides a better rate than $1/\sqrt{T}$ and sparse bounds any bound where some dependence on $d$ has been replaced with $\|\theta\|_0$. Our analysis starts from a careful study of the finite case $\Theta = \{\theta_1, \ldots, \theta_K\}$. We consider then online averaging algorithms on adaptive finite discretization grids that achieve sparse oracle bounds on $\Theta = \mathcal{B}_1 = \{\theta \in \mathbb{R}^d : \|\theta\|_1 \leqslant 1\}$.

**First contribution: fast-rate high probability quantile bound (finite $\Theta$, adversarial data)**  The case of finite reference set $\Theta = \{\theta_1, \ldots, \theta_K\}$ corresponds to the setting of prediction with expert advice (see Section 2.2 or [5]) where a learner makes sequential predictions over a series of rounds with the help of $K$ experts. Hedge introduced by [19] and [26] achieves the rate $R_T \leqslant \mathcal{O}(\sqrt{(\ln K)/T})$. The latter is optimal for general convex loss functions but better performance can be obtained in favorable scenarios. The rate $R_T \leqslant \mathcal{O}((\ln K)/T)$ is for instance obtained for strongly convex loss functions in [28]. Another improvement (see [16] and references therein) is devoted to quantile bounds, i.e. bounds on $\mathbb{E}_{k \sim \pi}[R_T(\theta_k)]$ for any probability distribution $\pi \in \Delta_K$[3]. The latter improve the dependence on the number of experts from $\ln K$ to the Kullback divergence $\mathcal{K}(\pi, \hat{\pi}_0)$ for any prior $\hat{\pi}_0$. They are smaller whenever many experts perform well or when a good prior knowledge is available. Squint [16] achieves a fast-rate quantile bound for adversarial data. Such a bound is obtained in high-probability by [20] but it suffers an additional gap term.

In Section 2, we extend the analysis of [16] to remove the gap term of [20]. We introduce a weak version of exp-concavity; see Assumption (A2). It depends on a parameter $\beta \in [0,1]$ which goes from $\beta = 0$ for general convex loss functions to $\beta = 1$ for exp-concavity. We show in Theorem 2.1 that BOA [28] and Squint [16] achieve a fast rate quantile bound with high probability: i.e. $\mathbb{E}_\pi[R_T(\theta_k)] \leqslant \tilde{\mathcal{O}}((\mathcal{K}(\pi, \hat{\pi}_0)/T)^{1/(2-\beta)})$.

**Second contribution: efficient sparse oracle bound ($\Theta = \mathcal{B}_1$, adversarial data)**  The extension from finite reference sets to convex sets is natural. The seminal paper [15] introduced the Exponentiated Gradient algorithm (EG), a version of Hedge using the sub-gradients of the loss functions. The latter guarantees $R_T \leqslant \mathcal{O}(\sqrt{(\ln d)/T})$ for $\Theta = \mathcal{B}_1$ which is optimal for convex loss functions. Recently, fast rate $R_T \leqslant \tilde{\mathcal{O}}((d/T)^{1/(2-\beta)})$ are obtained by [17] under a slightly different assumption than (A2). Here our purpose is to improve the dependence on $d$ under the sparsity condition $\|\theta\|_0$ small. The literature on learning under sparsity with i.i.d. data is vast; we refer to [12] for a review. Yet, little work was done on sparsity bounds under adversarial data; see Table 1 for a summary. The papers [7; 18; 29] focus on providing sparse estimators $\hat{\theta}_t$ rather than sparse guarantees. More recent works [8; 14] consider sparse approximations of the sub-gradients. Though they also compare themselves with sparse parameters, they incur a bound larger than $\mathcal{O}(1/\sqrt{T})$ which is optimal in their setting. Fast rate sparse regret bounds involving $\|\theta\|_0$ were, up to our knowledge, only obtained through non-efficient (exponential time) procedures (see [10]). In Section 3.3, we provide an efficient algorithm BOA+ which satisfies the oracle inequality

$$R_T(\theta) \leqslant \tilde{\mathcal{O}}\left((\sqrt{d\|\theta\|_0}/T) \wedge (\sqrt{\|\theta\|_0}/T^{3/4})\right), \qquad \text{for any } \theta \in \mathcal{B}_1,$$

for strongly-convex loss functions ($\beta = 1$). The gain $\sqrt{\|\theta\|_0/d} \wedge \sqrt{\|\theta\|_0/T}$ compared with the usual rate $\tilde{\mathcal{O}}(d/T)$ is significant for sparse parameters $\theta$.

A crucial step of our analysis is an intermediate result which is interesting in its own. We define an efficient algorithm with input any finite grid $\Theta_0 \subset \mathcal{B}_1$. We provide in Theorem 3.2 a bound of the form $R_T(\theta) \leqslant \tilde{\mathcal{O}}(D(\theta, \Theta_0)/\sqrt{T})$ for a pseudo-metric $D$ and any $\theta \in \mathcal{B}_1$. We say that this bound is *accelerable* as the rate may decrease if $D(\theta, \Theta_0)$ decreases with $T$. In particular, it yields an oracle bound of the form $R_T(\theta) \leqslant \mathcal{O}(\|\theta\|_1/\sqrt{T})$.

| Procedure | Rate | Polynomial | Assumption | Sparsity setting |
|---|---|---|---|---|
| Kale et al. [8; 14] | $\text{Poly}(d)/\sqrt{T}$ | Yes | Convexity | Sparse observed gradients |
| [7; 18; 29] | $\sqrt{\frac{\ln d}{T}}$ or $\frac{d}{T}$ | Yes | (Strong) Convexity | Produce sparse estimators |
| SeqSEW [11] | $\frac{d_0 \ln d}{T}$ | No | Strong Convexity | Sparse bound |
| SABOA | $\sqrt{\frac{\ln d}{T}} \ \wedge\ \frac{\sqrt{d_0 d}}{T} \ln d$ | Yes | Strong Convexity | Sparse bound |

Table 1: Comparison of sequential optimization procedures in sparse adversarial environment.

**Third contribution: sparse regret bound under Łojasiewicz assumption** ($\Theta = \mathcal{B}_1$, **i.i.d. data**) In Section 3.4 we turn to a stochastic setting where the loss functions $\ell_1, \ldots, \ell_T$ are i.i.d.. This setting extends the regression one with random design to general loss functions. The classical Lasso procedure satisfies, in the regression setting for the quadratic risk ($\beta = 1$), $R_T(\theta) \leqslant \tilde{\mathcal{O}}(\|\theta\|_0/T)$ where $\theta$ is a sparse approximation of $\theta^* = \arg\min_{\theta \in \mathbb{R}^d} R_T(\theta)$, see [3]. Yet, few procedures satisfying sparse bounds are sequential; we can cite [1; 8; 9; 14; 23]. We compare in Table 2 their results and settings.

The first line of work [1; 9; 23] provides sparse rates of order $\tilde{\mathcal{O}}(\|\theta^*\|_0 \ln d/T)$. Their settings are close to the one of [3] but their methods differ; the one of [23] uses a $\ell_1$-penalized gradient descent whereas the one of [1] and [9] are based on restarting a subroutine centered around the current estimate on sessions of exponentially growing length. A common limitation of these works is that they do not provide oracle inequality. They only compete with the global optimum over $\mathbb{R}^d$ only, which is assumed to be (approximately in [1]) sparse with a known $\ell_1$-bound. In other words, they assume that the global optimum also realizes the approximation-estimation errors trade-off in (1). In order to avoid this restriction, our first objective is to obtain the sparse bounds $R_T(\theta^*(U)) \leqslant \tilde{\mathcal{O}}(\|\theta^*(U)\|_0/T)$ where $\theta^*(U) \in \arg\min_{\|\theta\|_1 \leqslant U} R_T(\theta)$ for any $U > 0$. For $U$ well chosen so that $\|\theta^*(U)\|_1 = U$, $\theta^*(U)$ is sparse and the approximation-estimation errors trade-off in (1) is achieved. We restrict to the case $U = 1$ suppressing the dependence on $U$ in $\theta^*$ for the ease of notation. We leave the adaptation in $U > 0$ for future research.

The second line of works [14; 8] considers sparse approximation of sub-gradients. Yet, they provide a sparse regret bound of order $\mathcal{O}(\|\theta^*\|_0^2 \ln d/T)$ where $\theta^*$ is the optimum in $\mathcal{B}_1$ when the loss functions are strongly convex. Our second objective is to relax the strong convexity assumption which is too restrictive in the sequential regression setting. Indeed, the usual restricted eigenvalues conditions on the Gram matrix cannot hold uniformly for small $t$'s. We work under Łojasiewicz's Assumption introduced by [32; 33]: There exist $\beta > 0$ and $\mu > 0$ such that for all $\theta \in \mathcal{B}_1$, there exists a minimizer $\theta^*$ of the risk over $\mathcal{B}_1$ satisfying

$$\mu \big\| \theta - \theta^* \big\|_2^2 \leqslant \mathbb{E}[\ell_t(\theta) - \ell_t(\theta^*)]^\beta .$$

The Łojasiewicz assumption depends on a parameter $\beta \in [0, 1]$ that ranges from general convex risk function ($\beta = 0$) to generalized strongly convex risk function ($\beta = 1$). In Theorem 3.4 we show that our new efficient procedure SABOA achieves a fast rate upper-bound on the average excess risk of order $\tilde{\mathcal{O}}((\|\theta^*\|_0 \ln(d)/T)^{1/(2-\beta)})$ when the optimal parameters have $\ell_1$-norm bounded by $1 - \gamma < 1$. Then we recover the optimal rate of [1; 9; 23] in a similar setting, when the global optimum is assumed to be sparse. When $\|\theta^*\|_1 = 1$, guaranteeing a good approximation-estimation trade-off in (1), the bound suffers an additional factor $\|\theta^*\|_0$. Notice that Łojasiewicz's Assumption (A3) allows multiple optima which is important when we are dealing with degenerated co-linear design (allowing zero eigenvalues in the covariance matrix). It is an open question whether the fast rate $\tilde{\mathcal{O}}((\|\theta^*\|_0^2 \ln(d)/T))$ is optimal for efficient $\mathcal{O}(dT)$-complex procedures such as SABOA under Łojasiewicz's Assumption.

**Outline of the paper** To summarize our contributions, we provide
 - the first high-probability quantile bound achieving a fast rate in Theorem 2.1;
 - an accelerable bound on $R_T(\theta)$ that is small whenever $\theta$ is close to a prior grid $\Theta_0$ (Thm. 3.2);
 - two efficient algorithms with sparse regret bounds in the adversarial setting with strongly convex loss functions (BOA+, Thm. 3.3) and in the i.i.d. setting (SABOA, Thm. 3.4). In the latter setting, the results are obtained under the Łojasiewicz's assumption. This generalizes the usual necessary conditions for obtaining sparse bounds that are too restrictive in our sequential setting.

| Procedure | Setting | Rate | Assumptions / Setting | Optimum over |
|---|---|---|---|---|
| Lasso [3] | B | $d_0 \ln d/T$ | Mutual Coherence | $\mathbb{R}^d$ |
| Kale et al. [8; 14] | S | $d_0^2 \ln d/T$ | Strong Convexity + Sparse Gradients | $\mathcal{B}_1$ |
| [1; 9; 23]+SABOA | S | $d_0 \ln d/T$ | Strong convexity or Łojasiewicz ($\beta = 1$) | $\mathbb{R}^d$ |
| SABOA | S | $d_0^2 \ln d/T$ | Łojasiewicz ($\beta = 1$) | $\mathcal{B}_1$ |

Table 2: Comparison of sequential (S) and batched (B) optimization procedures in i.i.d. environment.

## 2 Finite reference set

In this section, we focus on finite reference set $\Theta := \{\theta_1, \dots, \theta_K\} \subset \mathcal{B}_1$, including the setting of prediction with expert advice presented in Section 2.2. We consider the following assumptions on the loss functions:

(A1) *Convex Lipschitz*[4]: the loss functions $\ell_t$ are convex on $\mathcal{B}_1$ and there exists $G > 0$ such that $\left\| \nabla \ell_t(\theta) \right\|_\infty \leqslant G$ for all $t \geq 1$, $\theta \in \mathcal{B}_1$.

(A2) *Weak exp-concavity:* There exist $\alpha > 0$ and $\beta \in [0, 1]$ such that for all $t \geqslant 1$, $\theta_1, \theta_2 \in \mathcal{B}_1$, almost surely

$$\mathbb{E}_{t-1}\big[\ell_t(\theta_1) - \ell_t(\theta_2)\big] \leqslant \mathbb{E}_{t-1}\big[\nabla \ell_t(\theta_1)^\top (\theta_1 - \theta_2)\big] - \mathbb{E}_{t-1}\Big[\Big(\alpha\big(\nabla \ell_t(\theta_1)^\top (\theta_1 - \theta_2)\big)^2\Big)^{1/\beta}\Big].$$

For convex loss functions ($\ell_t$), Assumption (A2) is satisfied with $\beta = 0$ and $\alpha < G^{-2}$. Fast rates are obtained for $\beta > 0$. It is worth pointing out that Assumption (A2) is weak even in the strongest case $\beta = 1$. It is implied by several common assumptions such as:

– *Strong convexity of the risk*: under the boundedness of the gradients, assumption (A2) with $\alpha = \mu/(2G^2)$ is implied by the $\mu$-strong convexity of the risks ($\mathbb{E}_{t-1}[\ell_t]$), $t \geq 1$.

– *Exp-concavity of the loss*: Lemma 4.2, Hazan [13] states that (A2) with $\alpha \leqslant \frac{1}{4} \min\{\frac{1}{8G}, \kappa\}$ is implied by $\kappa$-exp-concavity of the loss functions $\ell_t$, $t \geq 1$. Our assumption is slightly weaker since it holds in conditional expectation.

### 2.1 Fast-rate quantile bound with high probability

For prediction with $K \geqslant 1$ expert advice, [28] showed that a fast rate $\mathcal{O}\big((\ln K)/T\big)$ can be obtained by the BOA algorithm under the LIST condition (i.e., Lipschitz and strongly convex loss functions). In this section, we show that Assumption (A2) is enough and we improve the dependence on the total number of experts with a quantile bound.

Our algorithm is described in Algorithm 1 and corresponds to a particular case of two algorithms: the Squint algorithm of [16] used with a discrete prior over a finite set of learning rates and the BOA algorithm of [28] where each expert is replicated multiple times with different constant learning rates. The proof (with the exact constants) is deferred to Appendix C.1.

**Theorem 2.1.** *Let $T \geqslant 1$. Assume (A1) and (A2). Apply Algorithm 1, parameter $E = 4G/3$ and initial weight vector $\widehat{\pi}_0 \in \Delta_K$. Then, for all $\pi \in \Delta_K$, with probability at least $1 - 2e^{-x}$, $x > 0$,*

$$\mathbb{E}_{k \sim \pi}\left[R_T(\theta_k)\right] \lesssim \left(\frac{\mathcal{K}(\pi, \widehat{\pi}_0) + \ln \ln(GT) + x}{\alpha T}\right)^{\frac{1}{2-\beta}},$$

*where $\mathcal{K}(\pi, \widehat{\pi}_0) := \sum_{k=1}^{K} \pi_k \ln(\pi_k/\widehat{\pi}_{k,0})$ is the Kullback-Leibler divergence.*

A fast rate of this type (without quantiles property) can be obtained in expectation by using Hedge for exp-concave loss functions. However, Theorem 2.1 is stronger. First, Assumption (A2) is weaker than the exp-concavity of the loss functions $\ell_t$ as it holds for absolute or quantile loss functions in a sufficiently regular regression setting. Second, the algorithm uses the so-called gradient trick; See [24]. Therefore, simultaneously with the fast rate $\mathcal{O}\big(T^{-1/(2-\beta)}\big)$ with respect to the experts $(\theta_k)$,

**Algorithm 1** Squint – BOA with multiple constant learning rates assigned to each parameter

---

**Parameters:** $\Theta_0 = \{\theta_1, \dots, \theta_K\} \subset \mathcal{B}_1$, $E > 0$ and $\widehat{\pi}_0 \in \Delta_K$.
**Initialization:** For $1 \leqslant i \leqslant \ln(ET^2)$, define $\eta_i := (e^i E)^{-1}$.
For each iteration $t = 1, \dots, T$ do:

- Choose $\widehat{\theta}_{t-1} = \sum_{k=1}^{K} \widehat{\pi}_{k,t-1} \theta_k$ and observe $\nabla \ell_t(\widehat{\theta}_{t-1})$,

- Update component-wise for all $1 \leqslant k \leqslant K$

$$\widehat{\pi}_{k,t} = \frac{\sum_{i=1}^{\ln(ET^2)} \eta_i e^{\eta_i \sum_{s=1}^{t}(r_{k,s} - \eta_i r_{k,s}^2)} \widehat{\pi}_{k,0}}{\sum_{i'=1}^{\ln(ET^2)} \mathbb{E}_{j \sim \widehat{\pi}_0}\left[\eta_{i'} e^{\eta_{i'} \sum_{s=1}^{t}(r_{j,s} - \eta_{i'} r_{j,s}^2)}\right]}, \qquad r_{k,s} = \nabla \ell_t(\widehat{\theta}_{s-1})^{\top}(\widehat{\theta}_{s-1} - \theta_k).$$

---

the algorithm achieves the slow rate $\mathcal{O}(1/\sqrt{T})$ with respect to any convex combination $\mathbb{E}_{k \sim \pi}[\theta_k]$ (similarly to EG). Finally, high-probability regret bounds as ours are not satisfied by Hedge (see [2]).

If the algorithm is run with a uniform prior $\widehat{\pi}_0 = (1/K, \dots, 1/K)$, Theorem 2.1 implies that for any subset $\Theta' \subseteq \Theta$

$$\max_{\theta \in \Theta'} R_T(\theta) \lesssim \left(\frac{\ln(K/\operatorname{Card}(\Theta')) + \ln\ln(GT)}{\alpha T}\right)^{\frac{1}{2-\beta}} \qquad \text{with high probability.}$$

Thanks to the quantile bounds, we pay the proportion of good experts $\ln(K/\operatorname{Card}(\Theta'))$ in the regret instead of the total number of experts $\ln(K)$. We refer to [16] for more interesting applications. Such quantile bounds on the risk were studied by Mehta [20, Section 7] in a batch i.i.d. setting (i.e., $\ell_t$ are i.i.d.). A standard online to batch conversion shows that Theorem 2.1 yields with high probability

$$\mathbb{E}_T\left[\ell_{T+1}(\bar{\theta}_T) - \mathbb{E}_{k \sim \pi}[\ell_{T+1}(\theta_k)]\right] \lesssim \left(\frac{\mathcal{K}(\pi, \widehat{\pi}_0) + \ln\ln(GT) + x}{\alpha T}\right)^{\frac{1}{2-\beta}}, \qquad \bar{\theta}_T = \frac{1}{T}\sum_{t=1}^{T} \widehat{\theta}_{t-1}.$$

This improves the bound obtained by [20] who suffers the additional gap

$$(e-1)\,\mathbb{E}_T\left[\mathbb{E}_{k \sim \pi}[\ell_{T+1}(\theta_k)] - \min_{\pi^* \in \Delta_K} \ell_{T+1}(\mathbb{E}_{j \sim \pi^*}[\theta_j])\right].$$

## 2.2 Prediction with expert advice

The framework of prediction with expert advice is widely considered in the literature (see [5] for an overview). We recall now this setting and how it can be included in our framework. At the beginning of each round $t$, a finite set of $K \geqslant 1$ experts predict $\boldsymbol{f}_t = (f_{1,t}, \dots, f_{K,t}) \in [0,1]^K$ from the history $\mathcal{F}_{t-1}$. The learner then chooses a weight vector $\theta_{t-1}$ in the simplex $\Delta_K$ and produces a prediction $\widehat{f}_t := \widehat{\theta}_{t-1}^{\top} \boldsymbol{f}_t \in \mathbb{R}$ as a convex combination of the experts. Its performance at time $t$ is evaluated by a loss function $g_t : \mathbb{R} \to \mathbb{R}$. The goal of the learner is to approach the performance of the best expert on a long run. This can be done by minimizing the average excess risk $R_{k,T} := \frac{1}{T}\sum_{t=1}^{T} \mathbb{E}_{t-1}[g_t(\widehat{f}_t)] - \mathbb{E}_{t-1}[g_t(f_{k,t})]$, with respect to all experts $k \in \{1, \dots, K\}$. This setting reduces to our framework with dimension $d = K$. Indeed, it suffices to choose the $K$-dimensional loss function $\ell_t : \theta \mapsto g_t(\theta^{\top} \boldsymbol{f}_t)$ and the canonical basis $\Theta := \{\theta \in \mathbb{R}_+^K : \|\theta\|_1 = 1, \|\theta\|_0 = 1\}$ in $\mathbb{R}^K$ as the reference set. Denoting by $\theta_k$ the $k$-th element of the canonical basis, we see that $\theta_k^{\top} \boldsymbol{f}_t = f_{k,t}$, so that $\ell_t(\theta_k) = g_t(f_{k,t})$. Therefore, $R_{k,T}$ matches our definition of $R_T(\theta_k)$ in Equation (1) and we get under the assumptions of Theorem 2.1 a bound of order:

$$\mathbb{E}_{k \sim \pi}\left[R_{k,T}\right] \lesssim \left(\frac{\mathcal{K}(\pi, \widehat{\pi}_0) + \ln\ln(GT) + x}{\alpha T}\right)^{\frac{1}{2-\beta}}.$$

An important point to note here is that though the parameters $\theta_k$ of the reference set are constant, this method can be used to compare the player with arbitrary strategies $f_{k,t}$ that may evolve over time and depend on recent data. We do not assume in this section that there is a single fixed expert $k^* \in \{1, \dots, K\}$ which is always the best, i.e., $\mathbb{E}_{t-1}[g_t(f_{k^*,t})] \leqslant \min_k \mathbb{E}_{t-1}[g_t(f_{k,t})]$. Hence, we cannot replace (A2) with the closely related Bernstein assumption (see Ass. (A2') or [17, Cond. 1]).

Actually one can reformulate Assumption (A2) on the one dimensional loss functions $g_t$ as follows: there exist $\alpha > 0$ and $\beta \in [0,1]$ such that for all $t \geqslant 1$, for all $0 \leqslant f_1, f_2 \leqslant 1$,

$$\mathbb{E}_{t-1}[g_t(f_1) - g_t(f_2)] \leqslant \mathbb{E}_{t-1}\left[g_t'(f_1)(f_1 - f_2)\right] - \mathbb{E}_{t-1}\left[\left(\alpha\big(g_t'(f_1)(f_1 - f_2)\big)^2\right)^{1/\beta}\right], \quad a.s.$$

It holds with $\alpha = \kappa/(2G^2)$ for $\kappa$-strongly convex risk $\mathbb{E}_{t-1}[g_t]$. For instance, the square loss $g_t = (\cdot - y_t)^2$ satisfies it with $\beta = 1$ and $\alpha = 1/8$.

# 3 Online optimization in the unit $\ell_1$-ball

The aim of this section is to extend the preceding results to the reference set $\Theta = \mathcal{B}_1$ instead of finite $\Theta = \{\theta_1, \dots, \theta_K\}$. A classical reduction from the expert advice setting to the $\ell_1$-ball is the so-called "gradient-trick". A direct analysis on BOA applied to $\Theta_0 = \{\theta \in \mathbb{R}^d : \|\theta\|_0 = 1, \|\theta\|_1 = 1\}$ the $2d$ corners of the $\ell_1$-ball suffers a slow rate $\mathcal{O}(1/\sqrt{T})$ on the average excess risk with respect to any $\theta \in \mathcal{B}_1$. The goal is to exhibit algorithms that go beyond $\mathcal{O}(1/\sqrt{T})$. In Section 3.1 we investigate non-adaptive discretization grids of the space that yield optimal upper-bounds but suffer exponential time complexity. In Section 3.2 we introduce a pseudo-metric in order to bound the regret of grids consisting of the $2d$ corners and some arbitrary fixed points. From this crucial step, we derive the adaptive points to add to the $2d$ corners in the adversarial case (Section 3.3) and in the i.i.d. case (Section 3.4) in order to obtain two efficient procedures (BOA+ and SABOA respectively) with sparse guarantees.

## 3.1 Warmup: fast rate by discretizing the space

As a warmup, we show how to use Theorem 2.1 in order to obtain fast rate on $R_T(\theta)$ for any $\theta \in \mathcal{B}_1$. Basically, if the parameter $\theta$ could be included into the grid $\Theta_0$, Theorem 2.1 would turn into a bound on the regret $R_T(\theta)$ with respect to $\theta$. However, this is not possible as we do not know $\theta$ in advance. A solution consists in approaching $\mathcal{B}_1$ with $\mathcal{B}_1(\varepsilon)$, a fixed finite $\varepsilon$-covering in $\ell_1$-norm of minimal cardinal so that $\mathrm{Card}(\mathcal{B}_1(\varepsilon)) \lesssim (1/\varepsilon)^d$. We obtain a nearly optimal regret for this procedure.

**Proposition 3.1.** *Let $T \geqslant 1$. Under Assumptions of Theorem 2.1, applying Algorithm 1 with grid $\Theta_0 = \mathcal{B}_1(T^{-2})$ and uniform prior $\widehat{\pi}_0$ over $\Delta_{\mathrm{Card}(\mathcal{B}_1(T^{-2}))}$ satisfies for all $\theta \in \mathcal{B}_1$*

$$R_T(\theta) \lesssim \left( \frac{d \ln T + \ln \ln(GT) + x}{\alpha T} \right)^{\frac{1}{2-\beta}} + \frac{G}{T^2}, \tag{2}$$

*with probability at least $1 - e^{-x}$, $x > 0$.*

*Proof.* Let $\varepsilon = 1/T^2$ and $\theta \in \mathcal{B}_1$ and $\tilde{\theta}$ be its $\varepsilon$-approximation in $\mathcal{B}_1(\varepsilon)$. The proof follows from Lipschitzness of the loss: $R_T(\theta) \leqslant R_T(\tilde{\theta}) + G\varepsilon$ and by applying Theorem 2.1 on $R_T(\tilde{\theta})$. □

One can improve $d$ to $\|\theta\|_0 \ln d$ by carefully choosing the prior $\widehat{\pi}_0$ as in [21]; see Appendix A for details. The obtained rate is optimal up to log-factors. However, the complexity of the discretization is prohibitive (of order $T^d$) and non realistic for practical purpose.

## 3.2 Oracle bound for arbitrary fixed discretization grid

Let $\Theta_0 \subset \mathcal{B}_1$ be a finite set. The aim of this Section is to study the regret of Algorithm 1 with respect to any $\theta \in \mathcal{B}_1$. Similarly to Proposition 3.1, the average excess risk may be bounded as

$$R_T(\theta) \lesssim \left( \frac{\ln \mathrm{Card}(\Theta_0) + \ln \ln T + x}{\alpha T} \right)^{\frac{1}{2-\beta}} + G\|\theta' - \theta\|_1, \tag{3}$$

for any $\theta' \in \Theta_0$. We say that a regret bound is *accelerable* if it provides a fast rate except a term depending on the distance with the grid (i.e., the term in $\|\theta' - \theta\|_1$ in (3)) that decreases with $T$. This property will be crucial in obtaining fast rates by adapting the grid $\Theta_0$ sequentially. The regret bound (3) is not accelerable due to the second term that is constant. In order to find an accelerable regret bound, we introduce the notion of *averaging accelerability*, a pseudo-metric that replaces the $\ell_1$-norm in (3). We give the intuition behind this notion in the sketch of the proof of Theorem 3.2.

**Definition 3.1** (Averaging accelerability). *For any $\theta, \theta' \in \mathcal{B}_1$, we define*

$$D(\theta, \theta') := \min \left\{ 0 \leqslant \pi \leqslant 1 : \|\theta - (1 - \pi)\theta'\|_1 \leqslant \pi \right\}.$$

This averaging accelerability has several nice properties. In Appendix B, we provide a few concrete upper-bounds in terms of classical distances. For instance, Lemma B.1 provides the upper-bound $D(\theta, \theta') \leqslant \|\theta - \theta'\|_1 / (1 - \|\theta'\|_1 \wedge \|\theta\|_1)$. We are now ready to state our regret bound, when Algorithm 1 is applied with an arbitrary approximation grid $\Theta_0$.

**Theorem 3.2.** *Let $\Theta_0 \subset \mathcal{B}_1$ such that $\{\theta : \|\theta\|_1 = 1, \|\theta\|_0 = 1\} \subseteq \Theta_0$. Let Assumption (A1) and (A2) be satisfied. Then, Algorithm 1 applied with uniform prior $\widehat{\pi}_0$ over the elements of $\Theta_0$ and $E = 8G/3$, satisfies with probability $1 - e^{-x}$, $x > 0$,*

$$R_T(\theta) \lesssim \left(\frac{a}{\alpha T}\right)^{\frac{1}{2-\beta}} + GD(\theta, \Theta_0)\sqrt{\frac{a}{T}} + \frac{aG}{T}, \qquad \theta \in \mathcal{B}_1,$$

*where $a = \ln \operatorname{Card}(\Theta_0) + \ln\ln(GT) + x$ and $D(\theta, \Theta_0) := \min_{\theta' \in \Theta_0} D(\theta, \theta')$.*

*Sketch of proof.* The complete proof can be found in Appendix C.2. We give here the high-level ideas. Let $\theta' \in \Theta_0$ be a point in the grid $\Theta_0$ minimizing $D(\theta, \theta')$. Then one can decompose $\theta = (1 - \varepsilon)\theta' + \varepsilon \theta''$ for a unique point $\|\theta''\|_1 = 1$ and $\varepsilon := D(\theta, \theta')$. See Appendix C.2 for details. The regret bound can be decomposed into two terms:

- The first term quantifies the cost of picking the correct $\theta' \in \Theta_0$, bounded using Theorem 2.1;
- The second one is the cost of learning $\theta'' \in \mathcal{B}_1$ rescaled by $\varepsilon$. Using a classical slow-rate bound in $\mathcal{B}_1$, it is of order $\mathcal{O}(1/\sqrt{T})$.

The average excess risk $R_T(\theta)$ is thus of order

$$(1-\varepsilon)\underbrace{R_T(\theta')}_{\text{Thm 2.1}} + \varepsilon \underbrace{R_T(\theta'')}_{G\sqrt{\ln(\operatorname{Card}\Theta_0))/T}} \lesssim \left(\frac{\ln\operatorname{Card}(\Theta_0) + \ln\ln(GT) + x}{\alpha T}\right)^{\frac{1}{2-\beta}} + \varepsilon G\sqrt{\frac{\ln\operatorname{Card}(\Theta_0)}{T}}.$$

$\square$

Note that the bound of Theorem 3.2 is *accelerable* as its second term vanishes to zero on the contrary to Inequality (3). Theorem 3.2 provides an upper-bound which may improve the rate $\mathcal{O}(1/\sqrt{T})$ if the distance $D(\theta, \Theta_0)$ is small enough. By using the properties of the averaging accelerability (see Lemma B.1 in Appendix B), Theorem 3.2 provides some interesting properties of the rate in terms of $\ell_1$ distance. By including 0 into the grid $\Theta_0$, we get an oracle-bound of order $\mathcal{O}(\|\theta\|_1/\sqrt{T})$ for any $\theta \in \mathcal{B}_1$. Moreover a bound of order $R_T(\theta) \leqslant \mathcal{O}(\|\theta - \theta_k\|_1/(\gamma\sqrt{T}))$ is obtained for all $\theta_k \in \Theta_0$ and $\|\theta\|_1 \leqslant 1 - \gamma < 1$.

It is worth pointing out that the bound on the gradient $G$ can be substituted with the average gradient observed by the learner. The constant $G$ can be improved to the level of the noise in certain situations with vanishing gradients (see for instance Theorem 3 of [9]).

### 3.3 Fast-rate sparsity regret bound in the adversarial setting

In this section, we focus on the adversarial case where $\ell_t = \mathbb{E}_{t-1}[\ell_t]$ are $\mu$-strongly convex deterministic functions. In this case, Assumption (A2) is satisfied with $\beta = 1$ and $\alpha = \mu/(2G^2)$. Our algorithm, called BOA+, is defined as follows. For each doubling session $i \geqslant 0$, BOA+ chooses $\widehat{\theta}_t$ from time step $t_i = 2^i$ to $t_{i+1} - 1$ by restarting Algorithm 1 with uniform prior, parameter $E = 4G/3$ and updated discretization grid $\Theta_0$ indexed by $i$:

$$\Theta^{(i)} = \{[\theta_i^*]_k, k = 0, \ldots, d\} \cup \{\theta : \|\theta\|_1 = 2, \|\theta\|_0 = 1\},$$

where $\theta_i^* \in \arg\min_{\theta \in \mathcal{B}_1} \sum_{t=1}^{t_i - 1} \ell_t(\theta)$ is the empirical risk minimizer (or the leader) until time $t_i - 1$. The notation $[\cdot]_k$ denotes the hard-truncation with $k$ non-zero values. Remark that $\theta_i^*$ for $i = 1, 2, \ldots, \ln_2(T)$ can be efficiently computed approximately as the solution of a strongly convex optimization problem.

**Theorem 3.3.** *Assume the loss functions are $\mu$-strongly convex on $\mathcal{B}_2 := \{\theta \in \mathbb{R}^d : \|\theta\|_1 \leqslant 2\}$ with gradients bounded by $G$ in $\ell_\infty$-norm on $\mathcal{B}_2$. The average regret of BOA+ satisfies the oracle bound*

$$R_T(\theta) \leqslant \tilde{\mathcal{O}}\left(\min\left\{G\sqrt{\frac{\ln d}{T}}, \sqrt{\frac{\|\theta\|_0}{\mu}}\left(G\sqrt{\frac{\ln d}{T}}\right)^{\frac{3}{2}}, \frac{\sqrt{\|\theta\|_0 d}G^2 \ln d}{\mu T}\right\} + \frac{G^2 \ln d}{\mu T}\right), \quad \theta \in \mathcal{B}_1.$$

The proof is deferred to Appendix C.6. We emphasize that the bound can be rewritten as follows:

$$R_T(\theta) \leqslant \tilde{\mathcal{O}}\left(\min\left\{G\sqrt{\frac{\ln d}{T}}, \frac{\|\theta\|_0 G^2 \ln d}{\mu T}\right\}\min\left\{G\sqrt{\frac{\ln d}{T}}, \frac{d G^2 \ln d}{\mu T}\right\}\right)^{1/2}, \quad \theta \in \mathcal{B}_1 \setminus \{0\}.$$

It provides an intermediate rate between known optimal rates without sparsity $\mathcal{O}(\sqrt{\ln d/T})$ and $\tilde{\mathcal{O}}(d/T)$ and known optimal rates with sparsity $\mathcal{O}(\sqrt{\ln d/T})$ and (for non-efficient procedures only) $\tilde{\mathcal{O}}(\|\theta\|_0/T)$. If all $\theta_i^*$ are approximately $d_0$-sparse it is possible to achieve the optimal rate of order $\tilde{\mathcal{O}}(d_0/T)$, for any $\|\theta\|_0 \leqslant d_0$. We leave for future work whether it is possible to achieve it in general.

*Remark* 3.1. The strongly convex assumption on the loss functions can be relaxed (see Inequality (33) in the proof of Theorem 3.3) by assuming (A2) on $\mathcal{B}_2$ and that there exists $\mu > 0$ and $\beta \in [0,1]$ such that for all $t \geqslant 1$ and $\theta \in \mathcal{B}_1$

$$\mu\|\theta - \theta_t^*\|_2^2 \leqslant \left(\tfrac{1}{t}\sum_{s=1}^t(\ell_s(\theta) - \ell_s(\theta_t^*))\right)^\beta, \quad \text{where} \quad \theta_t^* \in \arg\min_{\theta\in\mathcal{B}_1}\sum_{s=1}^t\ell_s(\theta) \,. \qquad (4)$$

The rates will depend on $\beta$ as it is the case in Theorem 2.1. A specific interesting case is when $\|\theta_t^*\|_1 = 1$. Then $\theta_t^*$ is very likely to be sparse. Denote $S_t^*$ its support. Assumption (4) can be restricted in this case. Indeed any $\theta \in \mathcal{B}_1$ satisfies $\|\theta\|_1 \leqslant \|\theta_t^*\|_1$, which from Lemma 6 of [1] yields $\|\theta - \theta_t^*\|_1 \leqslant 2\|[\theta - \theta_t^*]_{S_t^*}\|_1$ where $[\theta]_S = (\theta_i \mathbb{1}_{i\in S})_{1\leqslant i\leqslant d}$. One can restrict Assumption (4) to hold on $S_t^*$ only. Such restricted conditions for $\beta = 1$ are common in the sparse learning literature and essentially necessary for the existence of efficient and optimal sparse procedures, see [31]. For obtaining regret bounds on BOA+, the restricted condition (4) with $\beta = 1$ should hold at any time $t \geq 1$, which is unlikely in the regression setting.

## 3.4   Fast-rate sparse excess risk bound in the i.i.d. setting

In this section, we assume the loss functions $\ell_t$ to be i.i.d. We provide an algorithm with fast-rate sparsity risk-bound on $\mathcal{B}_1$ by regularly restarting Algorithm 1 with an updated discretization grid $\Theta_0$ approaching the set of minimizers $\Theta^* := \arg\min_{\theta\in\mathcal{B}_1}\mathbb{E}[\ell_t(\theta)]$.

In the i.i.d. setting, a close inspection of the proof of Theorem 3.4 shows that we can replace Assumption (A2) with the Bernstein condition: there exists $\alpha' > 0$ and $\beta \in [0,1]$, such that for all $\theta \in \mathcal{B}_1$, all $\theta^* \in \Theta^*$ and all $t \geqslant 1$,

$$\alpha'\mathbb{E}\left[\left(\nabla\ell_t(\theta)^\top(\theta - \theta^*)\right)^2\right] \leqslant \mathbb{E}\left[\nabla\ell_t(\theta)^\top(\theta - \theta^*)\right]^\beta. \qquad (A2')$$

This fast-rate type stochastic condition is equivalent to the *central condition* (see [25, Condition 5.2]) and was already considered to obtain faster rates of convergence for the regret (see [17, Condition 1]).

**The Łojasiewicz assumption**   In order to obtain sparse oracle inequalities we work under Łojasiewicz's Assumption (A3) which is a relaxed version of strong convexity of the risk.

(A3) *Łojasiewicz's inequality:* $(\ell_t)_{t\geqslant 1}$ is an i.i.d. sequence and there exist $\beta \in [0,1]$ and $0 < \mu \leqslant 1$ such that, for all $\theta \in \mathbb{R}^d$ with $\|\theta\|_1 \leqslant 1$, there exists $\theta^* \in \Theta^* \subseteq \mathcal{B}_1$ satisfying

$$\mu\|\theta - \theta^*\|_2^2 \leqslant \mathbb{E}[\ell_t(\theta) - \ell_t(\theta^*)]^\beta \,.$$

This assumption is fairly mild. It is indeed satisfied with $\beta = 0$ and $\mu = 1$ as soon as the loss function is convex. For $\beta = 1$, this assumption is implied by the strong convexity of the risk $\mathbb{E}[\ell_t]$. Our framework is more general because
- multiple optima are allowed, which seems to be new when combined with sparsity bounds. An exception is [21] that provides the optimal sparse rate under a low-rank Gram matrix setting for the non-efficient ES algorithm;
- on the contrary to [23] or [9], our framework does not compete with the minimizer $\theta^*$ over $\mathbb{R}^d$ with a known upper-bound on the $\ell_1$-norm $\|\theta^*\|_1$. We consider the minimizer over the $\ell_1$-ball $\mathcal{B}_1$ only. The latter is more likely to be sparse and Assumption (A3) only needs to hold over $\mathcal{B}_1$.

Assumption (A2) (or (A2')) and (A3) are strongly related. Assumption (A3) is more restrictive because it is design dependent in the regression setting; The constant $\mu$ corresponds to the smallest non-zero eigenvalue of the covariance matrix while $\alpha = 1/G^2$ for the square loss functions. If $\Theta^* = \{\theta^*\}$ is a singleton than Assumption (A3) implies Assumption (A2') with $\alpha' \geqslant \mu/G^2$.

**Algorithm and excess risk bound**   Our new procedure called SABOA is described in Algorithm 2. Again it starts from the accelerable bound provided in Theorem 3.2 which is small if one of the points in $\Theta_0$ is close to $\Theta^*$. As BOA+, SABOA restarts BOA by adding current estimators of $\Theta^*$ into an updated grid $\Theta_0$. The new points added to the grid are slightly different between the two algorithms. They are truncated versions of the average of past iterates $\widehat{\theta}_{t-1}$ for SABOA and of the leader for BOA+. Remark that restart schemes under Łojasiewicz's Assumption is natural and was already used by [22]. We get the following upper-bound on the average excess risk. The proof that computes the exact constants is postponed to Appendix C.7.

---

**Algorithm 2** SABOA – Sparse Acceleration of BOA

---

**Parameters:** $E > 0$

**Initialization:** $t_i = 2^i$ for $i \geqslant 0$,

For each session $i = 0, \ldots$ do:

- Define $\bar{\theta}^{(i-1)} := 0$ if $i = 0$ and $\bar{\theta}^{(i-1)} := 2^{-i+1} \sum_{t=t_{i-1}}^{t_i-1} \widehat{\theta}_{t-1}$ otherwise,
- Define $\Theta^{(i)}$ a set of hard-truncated and dilated soft-thresholded versions of $\bar{\theta}^{(i-1)}$ as in (45),
- Denote $K_i := \text{Card}(\Theta^{(i)}) + 2d \leqslant (i+1)(1 + \ln d) + 3d$,
- At time step $t_i$, restart Algorithm 1 in $\Delta_{K_i}$ with parameters $\Theta_0 := \Theta^{(i)} \cup \{\theta : \|\theta\|_1 = 1, \|\theta\|_0 = 1\}$ (denote by $\theta_1, \ldots, \theta_{K_i}$ its elements), $E > 0$ and uniform prior $\widehat{\pi}_0$.
  In other words, for time steps $t = t_i, \ldots, t_{i+1} - 1$:
  - Choose $\widehat{\theta}_{t-1} = \sum_{k=1}^{K_i} \widehat{\pi}_{k,t-1} \theta_k$ and observe $\nabla \ell_t(\widehat{\theta}_{t-1})$,
  - Define component-wise for all $1 \leqslant k \leqslant K_i$, denoting $\eta_j := (e^j E)^{-1}$,

$$\widehat{\pi}_{k,t} = \frac{\sum_{j=1}^{\ln(ET^2)} \eta_j e^{\eta_j \sum_{s=t_i}^{t} (r_{k,s} - \eta_j r_{k,s}^2)} \widehat{\pi}_{k,0}}{\sum_{j=1}^{\ln(ET^2)} \mathbb{E}_{k' \sim \widehat{\pi}_0} \left[ \eta_j e^{\eta_j \sum_{s=t_i}^{t} (r_{k',s} - \eta_j r_{k',s}^2)} \right]},$$

where $r_{k,s} = \nabla \ell_t(\widehat{\theta}_{s-1})^\top (\widehat{\theta}_{s-1} - \theta_k)$.

---

**Theorem 3.4.** *Under Assumptions (A1), (A2) and (A3), Algorithm 2 with $E = 4/3G \geqslant 1$ satisfies with probability at least $1 - e^{-x}$, $x > 0$, the average excess risk bound*

$$R_T(\theta^*) \lesssim \left( \frac{\ln d + \ln \ln(GT) + x}{T} \left( \frac{1}{\alpha} + \frac{G^2}{\mu} \left( d_0^2 \wedge \frac{d_0}{\gamma^2} \right) \right) \right)^{\frac{1}{2-\beta}},$$

*where $d_0 = \max_{\theta^* \in \Theta^*} \|\theta^*\|_0$ and $0 \leqslant \gamma \leqslant 1$ satisfies $\Theta^* \subseteq \mathcal{B}_{1-\gamma}$.*

We conclude with some important remarks about Theorem 3.4. First, we point out that SABOA adapts automatically to unknown parameters $\delta$, $\beta$, $\alpha$, $\mu$ and $d_0$ to fulfill the rate of Theorem 3.4.

*On the radius of L1 ball.* We provide the analysis into $\mathcal{B}_1$, the $\ell_1$-ball of radius $U = 1$ only. However, one might need to compare with points into $\mathcal{B}_1(U)$, the $\ell_1$-ball of radius $U > 0$, in order to obtain a good approximation-estimation trade-off. This can be done by rescaling the loss functions $\theta \in \mathcal{B}_1 \mapsto \ell_t(U\theta)$ and applying our results with $UG$, $U^2\mu$ and $\alpha$ under Assumptions (A1), (A2) and (A3) on $\mathcal{B}_1(U)$. The main rate of convergence of Theorem 3.4 is unchanged. The optimal choice of the radius, if it is not imposed by the application, is left for future research.

*Support recovery.* When all $\theta^* \in \Theta^*$ lie on the border of the $\ell_1$-ball, they are likely to be sparse. One can relax Assumption (A3) to hold in sup-norm and in a restricted version similar as done in the end of Remark 3.1. In this interesting setting, we could not avoid a factor $d_0^2$. The reason is that our sequential algorithm recovers the (largest) support of $\theta^*$ (see Configuration 3 of Figure 1) in a framework where the necessary (for the rate $\|\theta^*\|_0$) Irreprensatibility Condition [27] does not hold.

**Conclusion** In this paper, we show that BOA is an optimal online algorithm for aggregating experts under very weak conditions on the loss. Then we aggregate sparse versions of the leader (BOA+) or of the average of BOA's iterates (SABOA) in the adversarial or in the i.i.d. setting, respectively. Aggregating both achieves sparse fast-rates of convergence in any case. These rates are deteriorated compared with the ideal one $\tilde{\mathcal{O}}\big((\|\theta\|_0/T)^{1/(2-\beta)}\big)$ that requires restrictive assumption for efficient algorithm. Our main condition (A3) is weaker and more realistic than the usual ones when seeking for sequential sparse rate bounds for any $t \geq 1$.

## Footnotes

[1]The average excess risk $R_T(\theta)$ generalizes the average regret more commonly used in the online learning literature by considering the Dirac masses on $\{\ell_t\}$ as conditional distributions so that $\ell_t = \mathbb{E}_{t-1}[\ell_t]$, $t \geq 1$.

[2]Throughout the paper $\lesssim$ denotes an approximate inequality which holds up to universal constants and $\tilde{\mathcal{O}}$ denotes an asymptotic inequality up to logarithmic terms.

[3]Here and subsequently, $\Delta_K := \{\pi \in [0,1]^K; \|\pi\|_1 = 1\}$ denotes the simplex of dimension $K \geq 1$.

[4]Throughout the paper, we assume that the Lipschitz constant $G$ in (A1) is known. It can be calibrated online with standard tricks such as the doubling trick (see [6] for instance) under sub-Gaussian conditions.

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
