[Supplementary Material]

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

# Supplementary material

## A  Sparse oracle inequality by discretizing the space

Inspired by the work of [21], one can improve $d$ to $\|\theta\|_0 \ln d$ in Proposition 3.1 by carefully choosing the prior $\widehat{\pi}_0$. To do so, we cover $\mathcal{B}_1$ by the subspaces

$$\mathcal{B}_1^\tau := \left\{ \theta \in \mathcal{B}_1 : \forall i \quad \tau_i = 0 \Rightarrow \theta_i = 0 \right\},$$

where $\tau \in \{0,1\}^d$ denotes a sparsity pattern which determines the non-zero components of $\theta \in \mathcal{B}_1^\tau$. For each sparsity pattern $\tau \in \{0,1\}^d$, the subspace $\mathcal{B}_1^\tau$ can be approximated in $\ell_1$-norm by an $\varepsilon$-cover $\mathcal{B}_1^\tau(\varepsilon)$ of size $\varepsilon^{-\|\tau\|_0}$. In order to obtain the optimal rate of convergence, we apply Algorithm 1 with $\Theta_0 = \cup_{\tau \in \{0,1\}^d} \mathcal{B}_1^\tau(\varepsilon)$ with a non-uniform prior $\widehat{\pi}_0$. The latter penalizes non-sparse $\tau$ to reflect their respective complexities. We assign to any $\theta \in \mathcal{B}_1^\tau(\varepsilon)$ the prior, depending on $\tau \in \{0,1\}^d$,

$$\widehat{\pi}_{\tau,0} = \left( \#\mathcal{B}_1^\tau(\varepsilon)(d+1)\binom{d}{d_0} \right)^{-1} \approx \frac{\varepsilon^{d_0}}{(d+1)\binom{d}{d_0}} \qquad \text{where} \quad d_0 = \|\tau\|_0 .$$

Note that the sum $\widehat{\pi}_{\tau,0}$ over $\theta \in \mathcal{B}_1^\tau(\varepsilon)$ and $\tau \in \{0,1\}^d$ is one. Therefore, Theorem 2.1 yields

$$R_T(\theta) \lesssim \left( \frac{\|\theta\|_0 \ln(dT/\|\theta\|_0) + x}{\alpha T} \right)^{\frac{1}{2-\beta}} + \frac{\|\theta\|_0 G}{T^2} , \tag{5}$$

by noting that $\binom{d}{\|\theta\|_0} \leqslant d^{\|\theta\|_0}$ and choosing $\varepsilon = \|\theta\|_0/T^2$. Similar optimal oracle inequalities for mixing arbitrary regressions functions are obtained by Yang [30] and Catoni [4].

## B  Properties of the averaging accelerability

In this appendix, we give a geometric interpretation of the *averaging accelerability* defined in Definition (3.1). We also provide several properties in terms of classical distances.

**Geometric insight**  Let $\theta \in \mathcal{B}_1$ be some unknown parameter and $\theta' \in \mathcal{B}_1$ a point approximating $\theta$. Let us define $\theta'' \in \mathcal{B}_1$ the unique point satisfying

$$\|\theta''\|_1 = 1 \qquad \text{and} \qquad \theta'' = \lambda(\theta - \theta') + \theta' \tag{6}$$

for some $\lambda \geq 1$. From this definition, we immediately derive that

$$\left\| \theta - \left( 1 - \frac{1}{\lambda} \right)\theta' \right\|_1 = \frac{\|\theta''\|_1}{\lambda} = \frac{1}{\lambda}$$

Therefore from Definition 3.1, we have $D(\theta, \theta') \leqslant \frac{1}{\lambda}$. Actually, this is an equality and we can write

$$D(\theta, \theta') = \max \left\{ \lambda \geq 1 : \|\lambda(\theta - \theta') + \theta'\|_1 \leqslant 1 \right\}^{-1} .$$

As the maximum is achieved, the averaging accelerability corresponds to the inverse of $\lambda$ in the definition (6) of the extrapolation point $\theta''$.

Figure 1: Averaging accelerability for 3 different configurations.

Figure 1 pictures several configurations of $\theta'$ and $\theta$ that lead to different averaging accelerability. The further $\theta''$ is from $\theta$, the smaller is $D(\theta, \theta')$ and the smaller is the averaging accelerability. When

$D(\theta, \theta') = 1/\lambda = 1$, then $\theta = \theta''$ and our regret bound does not improve the classic slow-rate $\mathcal{O}(1/\sqrt{T})$. That typically happens when $\|\theta\|_1 = 1$, as in the second configuration in Figure 1. In this case, a possible solution is to consider a larger ball (for instance of radius 2 instead of 1). This approach was considered in [9], see Figure 1 there. Another solution is to remark that even when $\|\theta\|_1 = 1$, the procedure is still accelerable ($D(\theta, \theta') < 1$) if the approximation $\theta'$ satisfies the same constraints than $\theta$ (see the third configuration in Figure 1 where $\theta''$ and $\theta$ are on the same edge of the ball). We make this statement more precise in the following subsections.

## B.1 The averaging accelerability in terms of classical distances

We provide in the next Lemmas a few concrete upper-bounds in terms of classical distances. The proofs are respectively postponed to Appendices C.3 to C.5. The first Lemma, states that the averaging accelerability can be upper-bounded with the $\ell_1$-distance.

**Lemma B.1.** *We have for any $\theta, \theta' \in \mathcal{B}_1$*

$$D(\theta, \theta') \leqslant \frac{\|\theta - \theta'\|_1}{\|\theta - \theta'\|_1 + 1 - \|\theta\|_1} \ .$$

The Lemma above has a main drawback. The averaging accelerability does not decrease with the $\ell_1$-distance if $\|\theta\|_1 = 1$. In this case, we thus need additional assumptions. The following Corollary upper-bounds the averaging accelerability in sup-norm as soon as a $\theta'$ has a support included into the one of $\theta$. This situation is represented in the third configuration of Figure 1.

**Lemma B.2.** *Let $\theta, \theta' \in \mathcal{B}_1$. Assume that $\|\theta'\|_1 \geqslant \|\theta\|_1$ and $\mathrm{sign}(\theta'_i) \in \{0, \mathrm{sign}(\theta_i)\}$ for all $1 \leqslant i \leqslant d$. Then,*

$$D(\theta, \theta') \leqslant 1 - \min_{1 \leqslant i \leqslant d} \frac{|\theta_i|}{|\theta'_i|} \leqslant \frac{\|\theta - \theta'\|_\infty}{\Delta} \ ,$$

*where $\Delta := \min_{i:\theta'_i \neq 0} |\theta_i|$.*

We want to emphasis here the two very different behavior of the averaging accelerability;

- in the case $\|\theta\|_1 < 1$: the averaging accelerability is proportional to $\|\theta - \theta'\|_1$.
- in the case $\|\theta\|_1 = 1$: the averaging accelerability may be smaller than 1 and lead to improved regret guarantees under extra assumptions: $\|\theta'\|_1 = 1$ and the support of $\theta'$ is included in the one of $\theta$. The relative gain is then proportional to $\|\theta\|_0 \|\theta - \theta'\|_\infty$.

## B.2 The averaging accelerability with an approximation in sup-norm in hand

Let us focus on the second case, where the averaging accelerability is controlled under the knowledge of the support of $\theta$. The second inequality in Lemma B.2 is interesting but yields an undesirable dependence on $\Delta := \min_{i:\theta_i \neq 0} |\theta_i|$, which can be arbitrarily small and which is at best of order $\|\theta\|_1/\|\theta\|_0$. Moreover, the recovery of the support of $\theta$ is a well studied difficult problem, see [27]. Thanks to the following Lemma, we ensure the averaging accelerability from any $\ell_\infty$-approximation $\theta'$ of $\theta$. We use a dilated soft-thresholding version of $\theta'$ as an approximation of $\theta$. For any $\varepsilon > 0$, let us introduce $S_\varepsilon$ the soft threshold operator so that $S_\varepsilon(x)_i = \mathrm{sign}(x_i)(|x_i| - \varepsilon)_+$ for all $1 \leqslant i \leqslant d$. The soft threshold operator is equivalent to the popular LASSO algorithm in the orthogonal design setting for the square loss. We couple the soft-thresholding with a dilatation that has the benefit of ensuring non thresholded coordinates faraway from zero. This allows to get rid of the unwanted factor $1/\Delta$ of the Lemma B.2. It is replaced with a factor $2\|\theta\|_0/\|\theta\|_1$ which corresponds (up to the factor 2) to the best possible scenario for the value of $\Delta$.

**Lemma B.3.** *Let $\theta, \theta' \in \mathcal{B}_1$ such that $\|\theta - \theta'\|_\infty \leqslant \varepsilon$ and $\|\theta\|_0 \leqslant d_0$. Then, define the dilated soft-threshold*

$$\tilde{\theta} := S_\varepsilon(\theta') \left( 1 + \frac{2d_0 \varepsilon}{\|S_\varepsilon(\theta')\|_1} \right) \wedge \frac{1}{\|S_\varepsilon(\theta')\|_1}$$

*where by convention $\tilde{\theta} = 0$ when $S_\varepsilon(\theta') = 0$. Then $\tilde{\theta}$ satisfies*

(i) $\|\tilde{\theta}\|_1 \geqslant \|\theta\|_1$ if $\tilde{\theta} \neq 0$
(ii) $\mathrm{sign}(\tilde{\theta}_i) \in \{0, \mathrm{sign}(\theta_i)\}$ for all $1 \leqslant i \leqslant d$
(iii) $D(\theta, \tilde{\theta}) \leqslant 2d_0 \varepsilon/\|\theta\|_1$ .

Performing this transformation requires the knowledge of the values of $\varepsilon$ and $d_0$ that are not observed. However, performing an exponential grid on $\varepsilon$ from $1/T$ to $U$ only harms the complexity by a factor $\ln(UT)$.

## C   Proofs

### C.1   Proof of Theorem 2.1

Algorithm 1 is a particular case of the Bernstein Online Aggregation algorithm (BOA) with fixed learning rates of [28][5]. We make more clear the connexion thereafter. We will start our proof with Theorem 3.2 of [28] that we recall now together with the definition of BOA. For each expert $j \in \mathcal{K}$ and each instance $t \geqslant 1$, from Equation (9) of [28], BOA assigns the weight

$$w_{j,t} := \frac{\exp\left(\eta_j \sum_{s=1}^{t} r_{j,s} - \eta_j^2 r_{j,s}^2\right) w_{j,0}}{\sum_{k \in \mathcal{K}} \exp\left(\eta_k \sum_{s=1}^{t} r_{k,s} - \eta_k^2 r_{k,s}^2\right) w_{k,0}} \tag{7}$$

where $(\eta_j)$ and $w_{j,0}$ are parameters of BOA which respectively correspond to the learning rates and the initial weight associated with each expert; and where $r_{j,t}$ are the instantaneous linearized regrets (denoted $\ell_{j,t}$ in [28]). In our case, $r_{j,t} = \nabla \ell_t(\widehat{\theta}_{t-1})^\top (\widehat{\theta}_{t-1} - \theta_j)$. Now, Theorem 3.2 of [28] states that for any distribution $\tilde{\pi}$ over the set of experts $1 \leqslant j \leqslant K$:

$$\sum_{t=1}^{T} \mathbb{E}_{j \sim \tilde{\pi}}[r_{j,t}] \leqslant \mathbb{E}_{j \sim \tilde{\pi}} \left[ \eta_j \sum_{t=1}^{T} r_{j,t}^2 + \frac{\ln(\tilde{\pi}_j / w_{j,0})}{\eta_j} + \frac{\ln(\mathbb{E}_{k \sim \pi_0}[\eta_k^{-1}] / \mathbb{E}_{k \sim \tilde{\pi}}[\eta_k^{-1}])}{\eta_j} \right]. \tag{8}$$

There are two main differences between BOA and Algorithm 1.

First, there is a subtle difference in the definition of the weights: we consider the weights $\pi_{j,t} = \eta_j w_{j,t} / \mathbb{E}_{k \sim \pi_t}[\eta_k w_{k,t}]$ instead of $w_{j,t}$. This only impacts the priors (that are multiplied by $\eta_j \mathbb{E}_{k \sim \pi_0}[\eta_k^{-1}]$) and allows to remove the last term in (8) as analyzed in the proof of [28, Theorem 3.2]. For this definition of weights, we thus get:

$$\sum_{t=1}^{T} \mathbb{E}_{j \sim \tilde{\pi}}[r_{j,t}] \leqslant \mathbb{E}_{j \sim \tilde{\pi}} \left[ \eta_j \sum_{t=1}^{T} r_{j,t}^2 + \frac{\ln(\tilde{\pi}_j / \pi_{j,0})}{\eta_j} \right]. \tag{9}$$

We refer the reader to the last equation of the proof of Theorem 3.2 of [28] for this inequality.

Second, in the original version of the BOA algorithm, each expert $\theta_k$ is assigned to a single learning rate $\eta_k$. In Algorithm 1 each parameter $\theta_k$ for $k = 1, \ldots, K$ is replicated several times, each replica being assigned a different learning rate $\eta_i = e^{-i} E^{-1}$ for $1 \leqslant i \leqslant \ln(ET^2)$. Algorithm 1 corresponds to applying BOA on this extended set where each expert $k$ has $\ln(ET^2)$ replica indexed by $i$ whose weights are cumulated into $\widehat{\pi}_{k,t}$. The initial weight $\widehat{\pi}_{k,0}$ of expert $k$ is uniformly distributed among its $\ln(ET^2)$ replica; each gets the initial weight $\tilde{\pi}_{j,0} = \widehat{\pi}_{k,0} / \ln(ET^2)$.

For each parameter $\theta_k, k \in \{1, \ldots, K\}$, let $1 \leqslant i_k \leqslant \ln(ET^2)$ be the index of a learning rate which will be chosen later by the analysis in order to optimize the final bound. Let $\pi$ be a distribution over the index set $\{1, \ldots, K\}$. We now apply Inequality (9) to a specific distribution $\tilde{\pi}$ on the replica. We choose $\tilde{\pi}$ so that it assigns all the mass $\pi_k$ on the replica $(k, i_k)$ and no mass on the replica $(k, i)$ for $i \neq i_k$. In other words, $\tilde{\pi}_j = \pi_k \mathbb{1}_{i=i_k}$. Then $\ln(\tilde{\pi}_j / \tilde{\pi}_{j,0}) = \ln(\pi_k / \widehat{\pi}_{k,0} \ln(ET^2))$ and Inequality (9) entails

$$\sum_{t=1}^{T} \mathbb{E}_{k \sim \pi}[r_{k,t}] \leqslant \mathbb{E}_{k \sim \pi} \left[ \underbrace{e^{-i_k} E^{-1}}_{:=\lambda_k} \sum_{t=1}^{T} r_{k,t}^2 + e^{i_k} E \left( \ln(\pi_k / \widehat{\pi}_{k,0}) + \ln \ln(ET^2) \right) \right]$$

$$= \mathbb{E}_{k \sim \pi} \left[ \lambda_k \sum_{t=1}^{T} r_{k,t}^2 + \frac{\ln(\pi_k / \widehat{\pi}_{k,0}) + \ln \ln(ET^2)}{\lambda_k} \right], \tag{10}$$

where we defined $\lambda_k := e^{-i_k} E^{-1}$. Now, by choosing $i_k$, this bound may be optimized with respect to any $\lambda_k$ of the form $e^{-i_k} E^{-1}$, with $1 \leqslant i_k \leqslant \ln(ET^2)$. To get the minimum over any $\lambda_k > 0$, we pay additional additive and multiplicative terms due to edge effects that we compute now. Fix $k > 0$ and define $V_k = \sum_{t=1}^{T} r_{k,t}^2$. The minimum is reached when both terms in (10) are equal. This yields the optimal choice $\lambda_k \approx (V_k / a_k)^{-1/2}$, where $a_k := \ln(\pi_k / \pi_{k,0}) + \ln \ln(ET^2)$. However, because of edge effects, this is only possible when $1/(ET)^2 \leqslant (V_k / a_k)^{-1/2} \leqslant 1/(Ee)$. We distinguish three cases:

- if $\sqrt{a_k/V_k} > 1/(eE)$: then, we choose $\lambda_k = 1/(eE)$, which yields:

$$\lambda_k V_k + \frac{a_k}{\lambda_k} \leqslant \frac{2a_k}{\lambda_k} = 2ea_k E \leqslant 6a_k E$$

- if $1/(ET)^2 \leqslant (V_k/a_k)^{-1/2} \leqslant 1/(Ee)$: then, we can choose $\lambda_k$ such that

$$\frac{\lambda_k}{\sqrt{e}} \leqslant (V_k/a_k)^{-1/2} \leqslant \sqrt{e}\lambda_k\,,$$

which entails $\lambda_k V_k + \frac{a_k}{\lambda_k} \leqslant 2\sqrt{e}\sqrt{a_k V_k} \leqslant 4\sqrt{a_k V_k}$

- if $\sqrt{a_k/V_k} < (ET)^{-2}$: then, the choice $\lambda_k = (ET)^{-2}$ gives

$$\lambda_k V_k + \frac{a_k}{\lambda_k} \leqslant 2\lambda_k V_k = \frac{2V_k}{E^2 T^2} \leqslant \frac{2}{T}\,,$$

because $r_{k,t}^2 \leqslant E^2$.

Putting the three cases together and plugging into Inequality (10) yields

$$\sum_{t=1}^{T} \mathbb{E}_{k\sim\pi}[r_{k,t}] \leqslant \mathbb{E}_{k\sim\pi}\left[4\sqrt{a_k V_k} + 6a_k E\right] + \frac{2}{T}. \tag{11}$$

We recall Young's inequality.

**Lemma C.1** (Young's inequality). *For all $a, b \geqslant 0$ and $p, q > 0$ such that $1/p + 1/q = 1$, then $ab \leqslant a^p/p + b^q/q$.*

Applying it, with $p = q = 2$, and $a = \sqrt{2\lambda_k V_k}$ and $b = \sqrt{8a_k/\lambda_k}$, we get $4\sqrt{a_k V_k} \leqslant \lambda_k V_k + 4a_k/\lambda_k$ for any $\lambda_k > 0$. Therefore, substituting into Inequality (11), for any distribution $\pi$ over $\{1, \ldots, K\}$, we have

$$\sum_{t=1}^{T} \mathbb{E}_{k\sim\pi}[r_{k,t}] \leqslant \mathbb{E}_{k\sim\pi}\left[\lambda_k V_k + \frac{4a_k}{\lambda_k} + 6a_k E\right] + \frac{2}{T}\,, \tag{12}$$

where we recall that $V_k = \sum_{t=1}^{T} r_{k,t}^2$ and $a_k = \ln(\pi_k/\pi_{k,0}) + \ln\ln(ET^2)$. For simplicity, from now on, we will denote $\mathbb{E}_{k\sim\pi}$ by $\mathbb{E}_\pi$. Using Theorem 4.1 of [28] for $\eta_{j,t} = \lambda_j$ independent of $t$, we obtain with probability $1 - e^{-x}$ and integrating with respect to $\pi$

$$\sum_{t=1}^{T} \mathbb{E}_{t-1}[\mathbb{E}_\pi[r_{k,t}]] \leqslant \sum_{t=1}^{T} \mathbb{E}_\pi[r_{k,t}] + \mathbb{E}_\pi\left[\lambda_k \sum_{t=1}^{T} r_{k,t}^2 + \frac{x}{\lambda_k}\right]$$

$$\overset{(12)}{\leqslant} \mathbb{E}_\pi\left[2\lambda_k \sum_{t=1}^{T} r_{k,t}^2 + \frac{x + 4a_k}{\lambda_k} + 6a_k E\right] + \frac{2}{T}. \tag{13}$$

To apply Assumption (A2), we need to transform the second order term (the sum of $r_{k,s}^2$ in the right-hand side) into a cumulative risk. This can be done using a Poissonian inequality for martingales (see for instance Theorem 9 of [9]): with probability at least $1 - e^{-x}$

$$\sum_{t=1}^{T} r_{k,t}^2 \leqslant 2\sum_{t=1}^{T} \mathbb{E}_{t-1}\left[r_{k,t}^2\right] + \frac{9}{4} E^2 x\,.$$

Substituting into the previous regret inequality, this yields for any $\lambda_k > 0$ and any distribution $\pi$ over $\{1, \ldots, K\}$

$$\sum_{t=1}^{T} \mathbb{E}_{t-1}\left[\mathbb{E}_\pi[r_{k,t}]\right] \leqslant \mathbb{E}_\pi\left[4\lambda_k \sum_{t=1}^{T} \mathbb{E}_{t-1}[r_{k,t}^2] + \frac{9}{2}\lambda_k E^2 x + \frac{4a_k + x}{\lambda_k} + 6a_k E\right] + \frac{2}{T}. \tag{14}$$

Now, we are ready to apply Assumption (A2) in order to cancel the sum in the right-hand side. Assumption (A2) ensures that for any time $t \geqslant 1$

$$\mathbb{E}_{t-1}\left[\ell_t(\widehat{\theta}_{t-1}) - \ell_t(\theta_k)\right] \leqslant \mathbb{E}_{t-1}[r_{k,t}] - \left(\alpha \mathbb{E}_{t-1}[r_{k,t}^2]\right)^{1/\beta}\,.$$

Therefore, summing over $t = 1, \ldots, T$ and using the preceding inequality with probability at least $1 - 2e^{-x}$

$$\mathbb{E}_\pi\left[\sum_{t=1}^T \mathbb{E}_{t-1}\big[\ell_t(\widehat{\theta}_{t-1}) - \ell_t(\theta_k)\big]\right] \leqslant \mathbb{E}_\pi\left[\sum_{t=1}^T \mathbb{E}_{t-1}[r_{k,t}] - \big(\alpha\mathbb{E}_{t-1}[r_{k,t}^2]\big)^{1/\beta}\right]$$

$$\leqslant \mathbb{E}_\pi\left[4\lambda_k \sum_{t=1}^T \mathbb{E}_{t-1}[r_{k,t}^2] - \sum_{t=1}^T \big(\alpha\mathbb{E}_{t-1}[r_{k,t}^2]\big)^{1/\beta} + \frac{9}{2}\lambda_k E^2 x + \frac{4a_k + x}{\lambda_k} + 6a_k E\right] + \frac{2}{T}. \tag{15}$$

Now, we use Young's inequality (see Lemma C.1) again to cancel the two sums in the right-hand side. Let $\gamma > 0$ to be fixed later by the analysis. Using $a = \mathbb{E}_{t-1}[r_{k,t}^2]/\gamma$, $b = \gamma$, $p = 1/\beta$, and $q = 1/(1 - \beta)$, it yields

$$\mathbb{E}_{t-1}[r_{k,t}^2] \leqslant \frac{\beta\big(\mathbb{E}_{t-1}[r_{k,t}^2]\big)^{1/\beta}}{\gamma^{1/\beta}} + (1 - \beta)\gamma^{1/(1-\beta)}.$$

Thus,

$$\lambda_k\mathbb{E}_{t-1}[r_{k,t}^2] \leqslant \frac{\lambda_k\beta\big(\mathbb{E}_{t-1}[r_{k,t}^2]\big)^{1/\beta}}{\gamma^{1/\beta}} + \lambda_k(1 - \beta)\gamma^{1/(1-\beta)}.$$

The choice $\gamma = (4\lambda_k\beta)^\beta/\alpha$ yields $4\lambda_k\beta/\gamma^{1/\beta} = \alpha^{1/\beta}$, which entails

$$4\lambda_k\mathbb{E}_{t-1}[r_{k,t}^2] - \big(\alpha\mathbb{E}_{t-1}[r_{k,t}^2]\big)^{1/\beta} \leqslant 4\lambda_k(1 - \beta)\gamma^{1/(1-\beta)}$$

$$= 4\lambda_k(1 - \beta)\Big(\frac{(4\lambda_k\beta)^\beta}{\alpha}\Big)^{1/(1-\beta)}$$

$$= 4(1 - \beta)(4\beta)^{\beta/(1-\beta)}\Big(\frac{\lambda_k}{\alpha}\Big)^{1/(1-\beta)}$$

$$\leqslant 4\Big(\frac{4\lambda_k}{\alpha}\Big)^{1/(1-\beta)}. \tag{16}$$

Summing over $t$ and substituting into Inequality (15), we get

$$\mathbb{E}_\pi\left[\sum_{t=1}^T \mathbb{E}_{t-1}\big[\ell_t(\widehat{\theta}_{t-1}) - \ell_t(\theta_k)\big]\right] \leqslant E_\pi\left[\underbrace{4\Big(\frac{4\lambda_k}{\alpha}\Big)^{1/(1-\beta)}T + \frac{4a_k + x}{\lambda_k}}_{=:R_k} + \frac{9}{2}\lambda_k E^2 x + 6a_k E\right] + \frac{2}{T}. \tag{17}$$

We optimize $\lambda_k$ by equalizing the two main terms of $R_k$:

$$4\Big(\frac{4\lambda_k}{\alpha}\Big)^{1/(1-\beta)}T = \frac{4a_k + x}{\lambda_k} \Leftrightarrow \lambda_k = \Big(\frac{4a_k + x}{4T}\Big)^{\frac{1-\beta}{2-\beta}}\Big(\frac{\alpha}{4}\Big)^{\frac{1}{2-\beta}}.$$

We express $R_k$ in termes of $\lambda_k$ using this identity

$$\frac{R_k}{T} = 2\frac{4a_k + x}{\lambda_k T} = 2\Big(\frac{4a_k + x}{\alpha T}\Big)^{\frac{1}{2-\beta}}4^{\frac{1-\beta}{2-\beta}} \leqslant 4\Big(\frac{16a_k + 4x}{\alpha T}\Big)^{\frac{1}{2-\beta}}.$$

The choice $\lambda_k = 1/(2E)$ would give

$$\frac{R_T}{T} \leqslant 4\Big(\frac{4\lambda_k}{\alpha}\Big)^{1/(1-\beta)} + \frac{4a_k + x}{T\lambda_k} \leqslant \frac{(4a_k + x)E}{T}.$$

So that we can assume $\lambda_k \leqslant 1/(2E)$ and

$$\frac{R_T}{T} \leqslant 4\Big(\frac{16a_k + 4x}{\alpha T}\Big)^{\frac{1}{2-\beta}} + \frac{(4a_k + x)E}{T}$$

Substituting into Inequality (17) and upper-bounding $\lambda_k E^2 \leqslant E/2$, gives

$$\frac{1}{T}\mathbb{E}_\pi\left[\sum_{t=1}^T \mathbb{E}_{t-1}\big[\ell_t(\widehat{\theta}_{t-1}) - \ell_t(\theta_k)\big]\right] \leqslant E_\pi\left[4\Big(\frac{16a_k + 4x}{\alpha T}\Big)^{\frac{1}{2-\beta}} + \frac{(10a_k + 4x)E}{T}\right] + \frac{2}{T^2}.$$

Since $x \mapsto x^{1/(2-\beta)}$ is concave, using Jensen's inequality and replacing $a_k = \ln(\pi_k/\pi_{k,0}) + \ln\ln(ET^2)$ entails,

$$E_\pi\left[\left(\frac{16a_k + 4x}{\alpha T}\right)^{\frac{1}{2-\beta}}\right] \lesssim \left(\frac{\mathbb{E}_\pi[a_k] + x}{\alpha T}\right)^{\frac{1}{2-\beta}} \overset{\text{(def of } a_k)}{=} \left(\frac{\mathcal{K}(\pi, \widehat{\pi}_0) + \ln\ln(ET^2) + x}{\alpha T}\right)^{\frac{1}{2-\beta}}$$

which concludes the proof.

## C.2 Proof of Theorem 3.2

We denote by $\theta_1, \ldots, \theta_K$ the elements of $\Theta_0$. We recall that we use a particular case of Algorithm 1. We can thus follow the proof of Theorem 2.1 and start from Inequality (11). We apply it to a Dirac distributions $\pi$ on $\{1, \ldots, K\}$. We get that for any $1 \leqslant k \leqslant K$, for any $\lambda_k > 0$,

$$\sum_{t=1}^T r_{k,t} \leqslant 4\sqrt{a\sum_{t=1}^T r_{k,t}^2} + 6aE + \frac{2}{T}. \tag{18}$$

where $a := \ln(K) + \ln\ln(ET^2)$ and where we remind the notation of the linearized instantaneous regret $r_{k,t} = \nabla\ell_t(\widehat{\theta}_{t-1})^\top(\widehat{\theta}_{t-1} - \theta_k)$ for $1 \leqslant k \leqslant K$.

Let $\theta^* \in \mathbb{R}^d$, let $\varepsilon := D(\theta^*, \Theta_0)$ and $k^* \in \{1 \leqslant k \leqslant K\}$ such that $\|\theta^* - (1-\varepsilon)\theta_{k^*}\|_1 \leqslant \varepsilon$. Then there exists $\tilde{\theta}$ with $\|\tilde{\theta}\|_1 \leqslant 1$ such that

$$\theta^* = (1-\varepsilon)\theta_{k^*} + \varepsilon\tilde{\theta}. \tag{19}$$

Since $\{\theta \in \mathcal{B}_1 : \|\theta\|_1 = 1, \|\theta\|_0 = 1\} \subset \Theta_0$, we can write $\tilde{\theta}$ as a combination of elements of $\Theta_0$. Hence, from (19), there exists a distribution $\pi = (\pi_1, \ldots, \pi_K) \in \Delta_K$ such that

$$\theta^* = \sum_{k=1}^K \pi_k\theta_k \quad \text{and} \quad 1 - \pi_{k^*} \leqslant \varepsilon.$$

Denoting $r_t := \nabla\ell_t(\widehat{\theta}_{t-1})^\top(\widehat{\theta}_{t-1} - \theta^*)$, we thus get

$$r_t := \nabla\ell_t(\widehat{\theta}_{t-1})^\top(\widehat{\theta}_{t-1} - \theta^*) = \nabla\ell_t(\widehat{\theta}_{t-1})^\top\left(\widehat{\theta}_{t-1} - \sum_{k=1}^K \pi_k\theta_k\right)$$

$$= \nabla\ell_t(\widehat{\theta}_{t-1})^\top(\widehat{\theta}_{t-1} - \mathbb{E}_{k\sim\pi}[\theta_k]) = \mathbb{E}_{k\sim\pi}\left[r_{k,t}\right],$$

and integrating Inequality (18) with respect to $\pi$, we obtain

$$\sum_{t=1}^T r_t \leqslant \mathbb{E}_{k\sim\pi}\left[4\sqrt{a\sum_{t=1}^T \left(\nabla\ell_t(\widehat{\theta}_{t-1})^\top(\widehat{\theta}_{t-1} - \theta^* + \theta^* - \theta_k)\right)^2}\right] + \frac{2}{T} + 6aE$$

$$\leqslant 4\sqrt{a\sum_{t=1}^T r_t^2} + 4\mathbb{E}_{k\sim\pi}\left[\sqrt{a\sum_{t=1}^T \left(\nabla\ell_t(\widehat{\theta}_{t-1})^\top(\theta^* - \theta_k)\right)^2}\right] + \frac{2}{T} + 6aE. \tag{20}$$

Let us upper bound the second term of the right hand side.

$$\mathbb{E}_{k\sim\pi}\left[\sqrt{\sum_{t=1}^T \left(\nabla\ell_t(\widehat{\theta}_{t-1})^\top(\theta^* - \theta_k)\right)^2}\right]$$

$$\leqslant \sqrt{\sum_{t=1}^T \|\nabla\ell_t(\widehat{\theta}_{t-1})\|_\infty^2 \sum_{k=1}^K \pi_k\|\theta^* - \theta_k\|_1}$$

$$\leqslant \sqrt{\sum_{t=1}^T \|\nabla\ell_t(\widehat{\theta}_{t-1})\|_\infty^2 \left(\pi_{k^*}\|\theta^* - \theta_{k^*}\|_1 + (1-\pi_{k^*})\max_{1\leqslant k\leqslant K}\|\theta^* - \theta_k\|_1\right)}$$

$$\leqslant \sqrt{\sum_{t=1}^T \|\nabla\ell_t(\widehat{\theta}_{t-1})\|_\infty^2 \left(\pi_{k^*}\|\theta^* - \theta_{k^*}\|_1 + 2(1-\pi_{k^*})\right)}, \tag{21}$$

where the last inequality is because $\|\theta^* - \theta_k\|_1 \leqslant \|\theta_k\|_1 + \|\theta^*\|_1 \leqslant 2$. We also have from the definition of $\theta^*$ (see before (19))

$$\|\theta^* - \theta_{k^*}\|_1 \leqslant \|\theta^* - (1-\varepsilon)\theta_{k^*} + \varepsilon\theta_{k^*}\|_1 \leqslant \|\theta^* - (1-\varepsilon)\theta_{k^*}\|_1 + \varepsilon\|\theta_{k^*}\|_1 \leqslant 2\varepsilon.$$

Therefore, substituting into (21) we get

$$\mathbb{E}_{k\sim\pi}\left[\sqrt{\sum_{t=1}^{T}\left(\nabla\ell_t(\widehat{\theta}_{t-1})^\top(\theta^* - \theta_k)\right)^2}\right] \leqslant 4\varepsilon\sqrt{\sum_{t=1}^{T}\|\nabla\ell_t(\widehat{\theta}_{t-1})\|_\infty^2} = 4\varepsilon\bar{G}_T\sqrt{T},$$

where $\bar{G}_T := \sqrt{\frac{1}{T}\sum_{t=1}^{T}\|\nabla\ell_t(\widehat{\theta}_{t-1})\|_\infty^2} \leqslant G$.

Therefore, substituting into Inequality (20), we have

$$\sum_{t=1}^{T}r_t \leqslant 4\sqrt{a\sum_{t=1}^{T}r_t^2 + 16\varepsilon\bar{G}_T\sqrt{aT} + \frac{2}{T} + 6aE}\,,$$

which yields by Young's inequality for any $\lambda > 0$

$$\sum_{t=1}^{T}r_t \leqslant \lambda\sum_{t=1}^{T}r_t^2 + \frac{4a}{\lambda} + \underbrace{16\varepsilon\bar{G}_T\sqrt{aT} + \frac{2}{T} + 6aE}_{=:z}\,. \tag{22}$$

Now, we recognize an inequality similar to Inequality (12). There only are a few technical differences which do not matter in the analysis: we consider here a Dirac distribution $\pi$ on the comparison parameter $\theta^*$ and we have some additional rest terms that we denote by $z := 16\varepsilon\bar{G}_T\sqrt{aT} + \frac{2}{T} + 6aE$ for simplicity. We can then follow the lines of the proof of Theorem 2.1 after Inequality (12)

$$\sum_{t=1}^{T}\mathbb{E}_{t-1}[r_t] \stackrel{\text{Thm 4.1 of [28]}}{\leqslant} \sum_{t=1}^{T}r_t + \lambda\sum_{t=1}^{T}r_t^2 + \frac{x}{\lambda}$$

$$\stackrel{(22)}{\leqslant} 2\lambda\sum_{t=1}^{T}r_t^2 + \frac{4a+x}{\lambda} + z$$

$$\stackrel{\text{Thm 9 of [9]}}{\leqslant} 4\lambda\sum_{t=1}^{T}\mathbb{E}_{t-1}[r_t^2] + \frac{4a+x}{\lambda} + \frac{9}{2}\lambda E^2 x + z. \tag{23}$$

Using Assumption (A2) then yields

$$\sum_{t=1}^{T}\mathbb{E}_{t-1}\big[\ell_t(\widehat{\theta}_{t-1}) - \ell_t(\theta^*)\big] \leqslant \sum_{t=1}^{T}\mathbb{E}_{t-1}[r_t] - \big(\alpha\mathbb{E}_{t-1}[r_t^2]\big)^{1/\beta}$$

$$\stackrel{(23)}{\leqslant} 4\lambda\sum_{t=1}^{T}\mathbb{E}_{t-1}[r_t^2] - \big(\alpha\mathbb{E}_{t-1}[r_t^2]\big)^{1/\beta} + \frac{4a+x}{\lambda} + \frac{9}{2}\lambda E^2 x + z$$

$$\stackrel{(16)}{\leqslant} 4\left(\frac{4\lambda}{\alpha}\right)^{1/(1-\beta)} + \frac{4a+x}{\lambda} + \frac{9}{2}\lambda E^2 x + z.$$

This yields an inequality similar to Inequality (17). Optimizing in $\lambda > 0$, as we did for Inequality (17) gives:

$$\lambda = \min\left\{\frac{1}{2E}, \left(\frac{4a+x}{4T}\right)^{\frac{1-\beta}{2-\beta}}\left(\frac{\alpha}{4}\right)^{\frac{1}{2-\beta}}\right\},$$

and

$$\frac{1}{T}\sum_{t=1}^{T}\mathbb{E}_{t-1}\big[\ell_t(\widehat{\theta}_{t-1}) - \ell_t(\theta^*)\big] \leqslant 4\left(\frac{16a+4x}{\alpha T}\right)^{\frac{1}{2-\beta}} + \frac{(4a+x)E}{T} + \frac{9Ex}{4T} + \frac{z}{T}\,.$$

where we recall that $a = \ln(K) + \ln\ln(ET^2)$, $z = 16\varepsilon\bar{G}_T\sqrt{aT} + \frac{2}{T} + 6aE$ and $\bar{G}_T :=$
$\sqrt{\frac{1}{T}\sum_{t=1}^{T}\|\nabla\ell_t(\widehat{\theta}_{t-1})\|_\infty^2} \leqslant G$. Replacing $z$ with its definition and simplifying yields

$$\frac{1}{T}\sum_{t=1}^{T}\mathbb{E}_{t-1}\big[\ell_t(\widehat{\theta}_{t-1}) - \ell_t(\theta^*)\big] \leqslant 4\left(\frac{16a+4x}{\alpha T}\right)^{\frac{1}{2-\beta}} + \frac{(10a+4x)E}{T} + 16\varepsilon\bar{G}_T\sqrt{\frac{a}{T}} + \frac{2}{T^2}. \quad (24)$$

Keeping the main terms only and replacing $\varepsilon := D(\theta, \Theta_0)$ concludes the proof.

### C.3 Proof of Lemma B.1

Let $\pi := \|\theta' - \theta\|_1 / (\|\theta' - \theta\|_1 + 1 - \|\theta\|_1)$. Then, thanks to the triangular inequality, we have

$$\big\|\theta - (1-\pi)\theta'\big\|_1 = \big\|(1-\pi)(\theta - \theta') + \pi\theta\big\|_1 \leqslant (1-\pi)\|\theta - \theta'\|_1 + \pi\|\theta\|_1$$
$$= \frac{(1 - \|\theta\|_1)\|\theta - \theta'\|_1 + \|\theta - \theta'\|_1\|\theta\|_1}{\|\theta - \theta'\|_1 + 1 - \|\theta\|_1} = \pi.$$

The Definition 3.1 of $D(\theta, \theta')$ concludes the proof.

### C.4 Proof of Lemma B.2

Denote $\pi := 1 - \min_{1\leqslant i\leqslant d} |\theta_i|/|\theta'_i|$. Then, for any $1 \leqslant i \leqslant d$, $|\theta_i| \geqslant (1-\pi)|\theta'_i|$. Because $\theta'_i$ and $\theta_i$ have same signs, this yields $|\theta_i - (1-\pi)\theta'_i| = |\theta_i| - (1-\pi)|\theta'_i|$ for all $1 \leqslant i \leqslant d$. Summing over $i = 1, \ldots, d$, entails

$$\big\|\theta - (1-\pi)\theta'\big\|_1 = \sum_{i=1}^{d}\Big|\theta_i - (1-\pi)\theta'_i\Big| = \sum_{i=1}^{d}|\theta_i| - (1-\pi)|\theta'_i|$$
$$= \|\theta\|_1 - (1-\pi)\|\theta'\|_1 \overset{\|\theta'\|_1\geqslant\|\theta\|_1}{\leqslant} \pi\|\theta\|_1 \leqslant \pi. \quad (25)$$

Therefore, the Definition 3.1 of $D(\theta, \theta')$ concludes the proof of the first inequality. Now, let $1 \leqslant i \leqslant d$, if $|\theta'_i| \leqslant |\theta_i|$ then $1 - |\theta_i|/|\theta'_i| \leqslant 0$ and the second inequality holds. Otherwise, we have

$$1 - \frac{|\theta_i|}{|\theta'_i|} = \frac{|\theta'_i| - |\theta_i|}{|\theta'_i|} \overset{|\theta'_i|\geqslant|\theta_i|}{=} \frac{|\theta'_i - \theta_i|}{|\theta'_i|} \overset{|\theta'_i|\geqslant|\theta_i|}{\leqslant} \frac{|\theta'_i - \theta_i|}{|\theta_i|} \leqslant \frac{\|\theta' - \theta\|_\infty}{\Delta},$$

which concludes the proof of the Lemma.

### C.5 Proof of Lemma B.3

Let $\theta, \theta' \in \mathcal{B}_1$ such that $\|\theta - \theta'\|_\infty \leqslant \varepsilon$. First, we check that $\tilde{\theta}$ satisfies the assumptions of Lemma B.2. Since $\|\theta' - \theta\|_\infty \leqslant \varepsilon$, for all coordinates $1 \leqslant i \leqslant d$, we have $S_\varepsilon(\theta')_i = 0$ or $\text{sign}(S_\varepsilon(\theta'))_i = \text{sign}(\theta_i)$. Therefore, $\text{sign}(\tilde{\theta}_i) = \text{sign}(S_\varepsilon(\theta')_i) \in \{0, \text{sign}(\theta_i)\}$. Furthermore,

$$\|S_\varepsilon(\theta')\|_1 \geqslant \sum_{i\in\text{Supp}(\theta)}\big|S_\varepsilon(\theta')_i\big| \geqslant \sum_{i\in\text{Supp}(\theta)}\big(|\theta'_i| - \varepsilon\big) \geqslant \sum_{i\in\text{Supp}(\theta)}\big(|\theta_i| - 2\varepsilon\big) \geqslant \|\theta\|_1 - 2d_0\varepsilon.$$

$$(26)$$

If $S_\varepsilon(\theta') = 0$, then $\|\tilde{\theta}\|_1 = 0$ and $\|\theta\|_1 \leqslant 2d_0\varepsilon$ so that $D(\theta, \tilde{\theta}) \leqslant 1 \leqslant 2d_0\varepsilon/\|\theta\|_1$. Therefore, we can assume from now that $S_\varepsilon(\theta') \neq 0$. By definition of $\tilde{\theta}$, Inequality (26) yields $\|\tilde{\theta}\|_1 = \big(\|S_\varepsilon(\theta')\|_1 + 2d_0\varepsilon\big) \wedge 1 \geqslant \|\theta\|_1$. Then $\tilde{\theta}$ satisfies the assumptions of Lemma B.2, which we can apply

$$D(\theta, \tilde{\theta}) \leqslant 1 - \min_{1\leqslant i\leqslant d}\frac{|\theta_i|}{|\theta'_i|} = \max_{i\in\text{Supp}(\tilde{\theta})}\frac{|\tilde{\theta}_i| - |\theta_i|}{|\tilde{\theta}_i|}. \quad (27)$$

We consider two cases:

- $\|S_\varepsilon(\theta')\|_1 \geqslant 1 - 2d_0\varepsilon$ in which case for $i \in \mathrm{Supp}(\tilde\theta)$

$$\tilde\theta_i = \frac{S_\varepsilon(\theta')_i}{\|S_\varepsilon(\theta')\|_1} = \frac{(|\theta_i'| - \varepsilon)\,\mathrm{sign}(\theta_i')}{\|S_\varepsilon(\theta')\|_1}$$

so that $|\tilde\theta_i| = (|\theta_i'| - \varepsilon)/\|S_\varepsilon(\theta')\|_1$ and upper-bounding $-|\theta_i| \leqslant -|\theta_i'| - \varepsilon$ we get

$$\frac{|\tilde\theta_i| - |\theta_i|}{|\tilde\theta_i|} = \frac{|\theta_i'| - \varepsilon - |\theta_i|\|S_\varepsilon(\theta')\|_1}{|\theta_i'| - \varepsilon} \leqslant \frac{|\theta_i'| - \varepsilon - (|\theta_i'| - \varepsilon)\|S_\varepsilon(\theta')\|_1}{|\theta_i'| - \varepsilon}$$

$$\leqslant 1 - \|S_\varepsilon(\theta')\|_1 \leqslant 2d_0\varepsilon \leqslant \frac{2d_0\varepsilon}{\|\theta\|_1}.$$

Substituting into Inequality (27) concludes this case.

- Otherwise $\|S_\varepsilon(\theta')\|_1 \leqslant 1 - 2d_0\varepsilon$ and for $i \in \mathrm{Supp}(\tilde\theta) = \mathrm{Supp}(S_\varepsilon(\theta'))$

$$|\tilde\theta_i| = |S_\varepsilon(\theta')_i|\Big(1 + \frac{2d_0\varepsilon}{\|S_\varepsilon(\theta')\|_1}\Big) = (|\theta_i'| - \varepsilon)\Big(1 + \frac{2d_0\varepsilon}{\|S_\varepsilon(\theta')\|_1}\Big),$$

which implies upper-bounding $-|\theta_i| \leqslant -|\theta_i'| - \varepsilon$,

$$\frac{|\tilde\theta_i| - |\theta_i|}{|\tilde\theta_i|} = \frac{(|\theta_i'| - \varepsilon)\Big(1 + \frac{2d_0\varepsilon}{\|S_\varepsilon(\theta')\|_1}\Big) - |\theta_i|}{(|\theta_i'| - \varepsilon)\Big(1 + \frac{2d_0\varepsilon}{\|S_\varepsilon(\theta')\|_1}\Big)}$$

$$\leqslant \frac{(|\theta_i'| - \varepsilon)\frac{2d_0\varepsilon}{\|S_\varepsilon(\theta')\|_1}}{(|\theta_i'| - \varepsilon)\Big(1 + \frac{2d_0\varepsilon}{\|S_\varepsilon(\theta')\|_1}\Big)}$$

$$= \frac{2d_0\varepsilon}{\|S_\varepsilon(\theta')\|_1 + 2d_0\varepsilon}$$

$$\leqslant \frac{2d_0\varepsilon}{\|\tilde\theta\|_1} \leqslant \frac{2d_0\varepsilon}{\|\theta\|_1}.$$

Substituting the obtained bounds in each cases into Inequality (27) concludes the proof.

### C.6 Proof of Theorem 3.3

We perform the proof for $\theta \in \mathcal{B}_{1/2}$ only. However, optimization on $\mathcal{B}_1$ can be obtained by renormalizing the loss functions considering $\ell_t(2\theta)$ instead of $\ell_t$. We leave this generalization to the reader. Let $\theta \in \mathcal{B}_{1/2}$ and denote $d_0 = \|\theta\|_0$. For simplicity, we also assume that $T = 2^I - 1$ and $d_0 \neq 0$.

*Part 1 ($\tilde{\mathcal{O}}(\sqrt{T})$ regret – logarithmic dependence on $d_0$ and $d$)* First, we prove the slow rate bound obtained by Algorithm 1. Let $i \geqslant 0$. Denote by $\theta_1, \ldots, \theta_{3d+1}$ the $3d + 1$ elements of $\Theta^{(i)}$. For any distribution $\pi \in \Delta_{3d+1}$ over $\Theta^{(i)}$, we have from Inequality (11):

$$\sum_{t=t_i}^{t_{i+1}-1} \mathbb{E}_{k\sim\pi}[r_{k,t}] \leqslant \mathbb{E}_{k\sim\pi}\left[4\sqrt{a_k V_k} + 6a_k E\right] + \frac{2}{T}. \tag{28}$$

where we recall $r_{k,t} \leqslant \nabla\ell_t(\widehat\theta_{t-1})^\top(\widehat\theta_{t-1} - \theta_k)$, $a_k := \ln(\pi_k/\pi_{k,0}) + \ln\ln(ET^2) \leqslant \ln(3d + 1) + \ln\ln(ET^2) =: a$ and $V_k \leqslant \sum_{t=t_i}^{t_{i+1}-1} r_{k,t}^2 \leqslant t_i G^2$. Let $\pi$ such that $\theta = \sum_{k=1}^{3d+1} \pi_k \theta_k$, then thanks to the convexity assumption on the loss functions, we have

$$\ell_t(\widehat\theta_{t-1}) - \ell_t(\theta) \leqslant \nabla\ell_t(\widehat\theta_{t-1})^\top(\widehat\theta_{t-1} - \theta_k) = \mathbb{E}_{k\sim\pi}[r_{k,t}].$$

Therefore, Inequality (28) yields

$$\sum_{t=t_i}^{t_{i+1}-1} \ell_t(\widehat\theta_{t-1}) - \ell_t(\theta) \leqslant 4G\sqrt{at_i} + 6Ea + \frac{2}{T}.$$

Summing over $i = 0, \ldots, j-1$ and substituting $t_i = 2^i$ we get for any $j \geqslant 1$:

$$\text{Reg}_j(\theta) := \sum_{t=1}^{t_j-1} \ell_t(\widehat{\theta}_{t-1}) - \ell_t(\theta) \leqslant 4G\sqrt{a}\sum_{i=0}^{j-1} 2^{i/2} + \underbrace{6Eaj + \frac{2j}{T}}_{=:z} \leqslant 10G\sqrt{a}2^{j/2} + z\,, \quad (29)$$

where we recall $a = \ln(3d+1) + \ln\ln(ET^2)$. In particular for $j = I \lesssim \ln T$ we obtain the first inequality stated by the theorem:

$$R_T(\theta) \leqslant \mathcal{O}\left(G\sqrt{\frac{\ln d + \ln\ln(ET)}{T}}\right).$$

*Part 2 ($\tilde{\mathcal{O}}(T^{1/4})$ regret – logarithmic dependence on d)* We prove by induction the second bound of the Theorem: that for some $c > 0$ and all $j \geqslant 0$, we have

$$\text{Reg}_j(\theta) \leqslant 48\frac{ad_0c^2G^2}{\mu} + j\frac{caG^2}{\mu} + 16\sqrt{5}c\sqrt{\frac{d_0\left(G\sqrt{a}\right)^3}{\mu}}\sum_{k=0}^{j} 2^{-\frac{3j}{4}}\,. \quad (30)$$

Indeed, decomposing the cumulative regret, we have

$$\text{Reg}_{j+1}(\theta) = \text{Reg}_j(\theta) + \sum_{t=t_j}^{t_{j+1}-1} \ell_t(\widehat{\theta}_{t-1}) - \ell_t(\theta)\,.$$

Note that Assumption (A2) is satisfied with $\beta = 1$, $\alpha = \mu/(2G^2)$ and without the expectation $\mathbb{E}_t$. It is worth pointing out that the transformation of the second-order term into a cumulative risk performed in (23) was not needed here since Assumption (A2) holds on the loss functions without the expectation $\mathbb{E}_t$. Therefore, the result of Theorem 3.2, that we can apply from time instance $t_j = 2^j$ to $t_{j+1} - 1$, holds almost surely with $x = 0$, $\beta = 1$ and $\alpha = \mu/(2G^2)$. We get that there exists some constant $c > 0$ such that

$$\sum_{t=t_j}^{t_{j+1}-1} \ell_t(\widehat{\theta}_{t-1}) - \ell_t(\theta) \leqslant cGD(\theta, [\theta_j^*]_{d_0})\sqrt{a2^j} + \frac{caG^2}{\mu}\,,$$

with $a = \ln(3d+1) + \ln\ln(ET^2)$. Replacing into the preceding inequality, it yields

$$\text{Reg}_{j+1}(\theta) \leqslant \text{Reg}_j(\theta) + cGD(\theta, [\theta_j^*]_{d_0})\sqrt{a2^j} + \frac{caG^2}{\mu} \quad (31)$$

Because $\theta \in \mathcal{B}_{1/2}$, we obtain from Lemma B.1

$$D\left(\theta, [\theta_j^*]_{d_0}\right) \overset{\text{(Lem. B.1)}}{\leqslant} 2\big\|\theta - [\theta_j^*]_{d_0}\big\|_1 \overset{\|\theta\|_0 = \|[\theta_j^*]_{d_0}\|_0 = d_0}{\leqslant} 2\sqrt{2d_0}\big\|\theta - [\theta_j^*]_{d_0}\big\|_2$$
$$\leqslant 2\sqrt{2d_0}\big(\big\|\theta - \theta_j^*\big\|_2 + \big\|\theta_j^* - [\theta_j^*]_{d_0}\big\|_2\big)\,.$$

By definition of the hard threshold, for any $\theta$ such that $\|\theta\|_0 = d_0$, we have

$$\big\|\theta_j^* - [\theta_j^*]_{d_0}\big\|_2 \leqslant \big\|\theta_j^* - \theta\big\|_2\,.$$

Therefore, plugging into the previous inequality

$$D\left(\theta, [\theta_j^*]_{d_0}\right) \leqslant 4\sqrt{2d_0}\big\|\theta - \theta_j^*\big\|_2\,. \quad (32)$$

But because the loss functions are $\mu$-strongly convex, the average loss over several rounds is also $\mu$-strongly convex. And since $\theta_j^* := \arg\min_{\theta \in \mathcal{B}_{1/2}} \sum_{t=1}^{t_j-1} \ell_t(\theta)$, we have for all $\theta \in \mathcal{B}_{1/2}$

$$\mu\big\|\theta - \theta_j^*\big\|_2^2 \leqslant \frac{1}{2^j - 1}\sum_{t=1}^{t_j-1} \ell_t(\theta) - \ell_t(\theta_j^*) \quad (33)$$
$$= \frac{\text{Reg}_j(\theta_j^*) - \text{Reg}_j(\theta)}{2^j - 1} \leqslant \frac{\text{Reg}_j(\theta_j^*) - \text{Reg}_j(\theta)}{2^{j-1}}\,.$$

Thus, from Inequality (32), we obtain

$$D\big(\theta, [\theta_j^*]_{d_0}\big) \leqslant 8\sqrt{\frac{d_0\big(\operatorname{Reg}_j(\theta_j^*) - \operatorname{Reg}_j(\theta)\big)}{\mu 2^j}} \, .$$

Plugging into Inequality (31) gives

$$\operatorname{Reg}_{j+1}(\theta) \leqslant \operatorname{Reg}_j(\theta) + 8cG\sqrt{\frac{ad_0}{\mu}\big(\operatorname{Reg}_j(\theta_j^*) - \operatorname{Reg}_j(\theta)\big)} + \frac{caG^2}{\mu} \, . \qquad (34)$$

We can upper-bound $\operatorname{Reg}_j(\theta_j^*)$ using Inequality (29). This entails

$$\operatorname{Reg}_{j+1}(\theta) \leqslant \operatorname{Reg}_j(\theta) + 8cG\sqrt{\frac{ad_0}{\mu}\big(10G\sqrt{a}2^{j/2} + z - \operatorname{Reg}_j(\theta)\big)} + \frac{caG^2}{\mu} \, .$$

Now we have an inequality of the form

$$\operatorname{Reg}_{j+1}(\theta) \leqslant \operatorname{Reg}_j(\theta) + x_1\sqrt{x_2 - \operatorname{Reg}_j(\theta)} + x_3$$

with $x_1 = 8cG\sqrt{ad_0/\mu}$, $x_2 = 10G\sqrt{a}2^{j/2} + z$ and $x_3 = (caG^2)/\mu$. If $\operatorname{Reg}_j(\theta) \geqslant 0$, $\operatorname{Reg}_{j+1}(\theta)$ is increased by at most $x_1\sqrt{x_2} + x_3$. Otherwise $\operatorname{Reg}_j(\theta) \leqslant 0$ and the right-hand side is at most $3x_1^2/4 + x_3$ (considering the maximum over $\operatorname{Reg}_j(\theta) \leqslant 0$). Therefore,

$$\operatorname{Reg}_{j+1}(\theta) \leqslant \max\big\{3x_1^2/4, (\operatorname{Reg}_j(\theta))_+ + x_1\sqrt{x_2}\big\} + x_3 \, .$$

$$= \max\left\{48\frac{ad_0c^2G^2}{\mu}, \ (\operatorname{Reg}_j(\theta))_+ + 16\sqrt{5}c\sqrt{\frac{d_0}{\mu}}\left(G\sqrt{\frac{a}{2^j}}\right)^{3/2}\right\} + \frac{caG^2}{\mu} \, . \qquad (35)$$

This concludes the induction, using the hypothesis (30). In particular, considering $j = I = \ln_2(T-1)$, we proved that

$$R_T(\theta) = \frac{\operatorname{Reg}_I(\theta)}{T} \leqslant \mathcal{O}\left(\sqrt{\frac{d_0}{\mu}}\left(G\sqrt{\frac{\ln d + \ln\ln(ET)}{T}}\right)^{\frac{3}{2}}\right) \, .$$

*Part 3. ($\tilde{\mathcal{O}}(1)$ regret – square root dependence on $d$)* Now, we prove a faster rate but at the price of a square root dependence in the total dimension $d$. The proof follows the same lines as the preceding part except that one changes the induction hypothesis and that one uses it to bound the regret of $\theta_j^*$. We prove by induction: there exists $c_0 > 0$ such that for any $\theta \in \mathcal{B}_{1/2}$

$$\operatorname{Reg}_j(\theta) \leqslant j\frac{ac_0\sqrt{\|\theta\|_0 d}G^2}{\mu T}$$

where $a = \ln(3d + 1) + \ln\ln(ET^2)$. We start from Inequality (34) obtained in Part 2:

$$\operatorname{Reg}_{j+1}(\theta) \leqslant \operatorname{Reg}_j(\theta) + 8cG\sqrt{\frac{ad_0}{\mu}\big(\operatorname{Reg}_j(\theta_j^*) - \operatorname{Reg}_j(\theta)\big)} + \frac{caG^2}{\mu} \, .$$

Now, instead of upper-bounding $\operatorname{Reg}_j(\theta_j^*)$ using Inequality (29), we use the induction hypothesis itself. Since $\theta_j^*$ is not necessarily sparse, we have

$$\operatorname{Reg}_j(\theta_j^*) \leqslant j\frac{ac_0 dG^2}{\mu} \, ,$$

which entails

$$\operatorname{Reg}_{j+1}(\theta) \leqslant \operatorname{Reg}_j(\theta) + 8cG\sqrt{\frac{ad_0}{\mu}\Big(j\frac{ac_0 dG^2}{\mu} - \operatorname{Reg}_j(\theta)\Big)} + \frac{caG^2}{\mu} \, .$$

We obtain a regret bound of the same form than in Part 2:

$$\mathrm{Reg}_{j+1}(\theta) \leqslant \mathrm{Reg}_j(\theta) + x_1\sqrt{x_2 - \mathrm{Reg}_j(\theta)} + x_3,$$

with $x_1 = 8cG\sqrt{ad_0/\mu}$, $x_2 = (jac_0dG^2)/\mu$ and $x_3 = caG^2/\mu$. Similarly to Inequality (35), we have

$$\mathrm{Reg}_{j+1}(\theta) \leqslant \max\left\{3x_1^2/4, (\mathrm{Reg}_j(\theta))_+ + x_1\sqrt{x_2}\right\} + x_3$$

$$= \max\left\{48\frac{ad_0c^2G^2}{\mu}, (\mathrm{Reg}_j(\theta))_+ + \frac{8c\sqrt{c_0}a\sqrt{d_0}dG^2}{\mu}\right\} + \frac{caG^2}{\mu}$$

$$\leqslant (\mathrm{Reg}_j(\theta))_+ + \frac{(49 + 8c\sqrt{c_0})a\sqrt{d_0}dG^2}{\mu}.$$

Choosing $c_0 > 0$ such that $49 + 8c\sqrt{c_0} \leqslant c_0$ concludes the induction. In particular, considering $j = I = \ln_2(T-1)$, we proved that

$$R_T(\theta) \leqslant \mathcal{O}\left(\frac{\sqrt{d_0d}G^2(\ln d + \ln\ln(ET))\ln T}{\mu T}\right).$$

## C.7   Proof of Theorem 3.4

We recall that $\Theta^* = \arg\min_{\theta\in\mathcal{B}_1}\mathbb{E}[\ell_t(\theta)]$. The idea of the proof is to show that at each session $i$, SABOA performs BOA by adding sparse estimators in $\Theta^{(i)}$ that are exponentially closer to $\Theta^*$.

Let $x > 0$. We prove by induction on $i \geqslant 0$ that with probability at least $1 - ie^{-x}$, there exists $\theta^* \in \Theta^*$ such that

$$D(\theta^*, \Theta^{(i)}) \leqslant \varepsilon 2^{-\tau i}, \tag{$\mathcal{H}_i$}$$

where $D$ is defined in Definition 3.1,

$$\varepsilon := \max_{\theta^*\in\Theta^*}\left((8\sqrt{a}G)^\beta \max\left\{\frac{2}{\alpha G^2}, \frac{8\|\theta^*\|_0}{\mu}\min\left\{\frac{8\|\theta^*\|_0}{\|\theta^*\|_1^2}, \frac{1}{(1-\|\theta^*\|_1)^2}\right\}\right\}\right)^{\frac{1}{2-\beta}}, \tag{36}$$

and $\tau = \frac{1}{2-\beta} - \frac{1}{2}$. Remark that $\theta^*$ in $(\mathcal{H}_i)$ depends on $i$ when $\Theta^*$ is not a singleton.

*Initialization.* For $i = 0$, by definition (see Algorithm 2), $\Theta^{(0)} := \{0\}$ and $D(\theta^*, \{0\}) \leqslant \|\theta^*\|_1 \leqslant 1$. The initialization thus holds true as soon as $\varepsilon > 1$.

*Induction step.* Let $i \geqslant 0$ and assume $(\mathcal{H}_i)$. We start from Theorem 3.2 (see Inequality (24) for the precise constants that we upper-bound here) that we apply for $t = t_{i-1}, \dots, t_i - 1$ and $\theta^* \in \Theta^*$ satisfying $(\mathcal{H}_i)$: with probability $1 - e^{-x}$

$$\frac{1}{2^{i-1}}\sum_{t=t_{i-1}}^{t_i-1}\mathbb{E}_{t-1}\left[\ell_t(\widehat{\theta}_{t-1}) - \ell_t(\theta^*)\right] \leqslant \frac{2\sqrt{a}GD(\theta^*, \Theta^{(i)})}{2^{(i-1)/2}} + 4\left(\frac{a}{\alpha 2^{i-1}}\right)^{\frac{1}{2-\beta}} + \frac{aE}{2^{i-1}} + \frac{2}{2^{2i-2}},$$

where for simplicity of notation we define $a := 16(1 + \ln(K_i)) + 16\ln\ln(ET^2) + 4x$, where $K_i := \mathrm{Card}(\Theta^{(i)}) + 2d$ denotes the number of experts used during the doubling session $i$, and where we used $t_i = t_{i-1} + 2^{i-1}$. Using $(\mathcal{H}_i)$ together with Jensen's inequality and recalling $\bar{\theta}^{(i)} := 2^{-i+1}\sum_{t=t_{i-1}}^{t_i-1}\widehat{\theta}_{t-1}$, we obtain

$$\mathbb{E}\left[\ell_t(\bar{\theta}^{(i)}) - \ell_t(\theta^*)\right] \leqslant 2\sqrt{2a}G\varepsilon 2^{-(\frac{1}{2}+\tau)i} + 4\left(\frac{a}{\alpha}\right)^{\frac{1}{2-\beta}}2^{-\frac{i}{2-\beta}} + aE2^{1-i} + 2^{3-2i}. \tag{37}$$

Now, we simplify this expression by showing that the last three terms of the right-hand side are negligible with respect to the first one. First, because $a \geqslant 16$ and $E \geqslant 1$, we have $16 \leqslant aE$ and thus $2^{3-2i} \leqslant aE2^{-1-i}$. Then, because $\varepsilon \geqslant \sqrt{a}$, $aE \leqslant \sqrt{a}E\varepsilon = \frac{4}{3}\sqrt{a}\varepsilon G$ and thus

$$aE2^{1-i} + 2^{3-2i} \leqslant \frac{3}{2}aE2^{-i} \leqslant 2\sqrt{a}\varepsilon G2^{-i} \overset{\tau\leqslant 1/2}{\leqslant} 2\sqrt{a}\varepsilon G2^{-(\frac{1}{2}+\tau)i}. \tag{38}$$

The second term is also dominated thanks to the definition of $\varepsilon$ in (36)

$$\varepsilon \overset{(36)}{\geqslant} \left(\frac{2(\sqrt{8a}G)^\beta}{\alpha G^2}\right)^{\frac{1}{2-\beta}} \quad \Rightarrow \quad 2\sqrt{2a}G\varepsilon \geqslant \left(\frac{16a}{\alpha}\right)^{\frac{1}{2-\beta}} \overset{0\leqslant\beta\leqslant1}{\geqslant} 4\left(\frac{a}{\alpha}\right)^{\frac{1}{2-\beta}}$$

and

$$\tau \overset{(36)}{=} \frac{1}{2-\beta} - \frac{1}{2} \quad \Rightarrow \quad \frac{1}{2-\beta} \geqslant \frac{1}{2} + \tau$$

which yields

$$4\left(\frac{a}{\alpha}\right)^{\frac{1}{2-\beta}} 2^{-\frac{i}{2-\beta}} \leqslant 2\sqrt{2a}G\varepsilon 2^{-(\frac{1}{2}+\tau)i} . \tag{39}$$

Thus replacing Inequalities (38) and (39) into Inequality (37) and upper-bounding $4\sqrt{2} + 2 \leqslant 8$, we get for any $\theta^* \in \Theta^*$

$$\mathbb{E}\left[\ell_t(\bar{\theta}^{(i)}) - \ell_t(\theta^*)\right] \leqslant 8\sqrt{a}G\varepsilon 2^{-(\frac{1}{2}+\tau)i} . \tag{40}$$

Using Assumption (A3), there exists at least one $\theta^* \in \Theta^*$ (which can be different from the preceding session), which satisfies

$$\left\|\bar{\theta}^{(i)} - \theta^*\right\|_\infty \leqslant \left\|\bar{\theta}^{(i)} - \theta^*\right\|_2 \overset{(40)+(A3)}{\leqslant} (8\sqrt{a}G\varepsilon)^{\frac{\beta}{2}} \mu^{-\frac{1}{2}} 2^{-(\frac{1}{2}+\tau)\frac{\beta}{2}i} =: \varepsilon' . \tag{41}$$

Now, we want to apply Lemma B.3 if $\|\theta^*\|_1$ is close to 1 and Lemma B.1 if $\|\theta^*\|_1 < 1$. In order to apply Lemma B.1, we consider hard-truncated estimators $[\bar{\theta}^{(i)}]_{\tilde{d}_0}$, canceling the $d - \tilde{d}_0$ smallest components of $\bar{\theta}^{(i)}$ for $\tilde{d}_0 \in \{1, \ldots, d\}$. For the (unknown) choice $\tilde{d}_0 = d_0$, since $\|[\bar{\theta}^{(i)}]_{d_0}\|_0 = \|\theta^*\|_0 = d_0$, we have $\|[\bar{\theta}^{(i)}]_{d_0} - \theta^*\|_0 \leqslant 2d_0$ and

$$\left\|[\bar{\theta}^{(i)}]_{d_0} - \theta^*\right\|_1 \leqslant \sqrt{2d_0}\left\|[\bar{\theta}^{(i)}]_{d_0} - \theta^*\right\|_2 \leqslant \sqrt{2d_0}\left(\left\|[\bar{\theta}^{(i)}]_{d_0} - \bar{\theta}^{(i)}\right\|_2 + \left\|\bar{\theta}^{(i)} - \theta^*\right\|_2\right)$$
$$\leqslant 2\sqrt{2d_0}\left\|\bar{\theta}^{(i)} - \theta^*\right\|_2 \leqslant 2\sqrt{2d_0}\varepsilon' .$$

Applying Lemma B.1, we get

$$D\left(\theta^*, [\bar{\theta}^{(i)}]_{d_0}\right) \leqslant \frac{\left\|[\bar{\theta}^{(i)}]_{d_0} - \theta^*\right\|_1}{1 - \|\theta^*\|_1} \leqslant \frac{2\sqrt{2d_0}\varepsilon'}{1 - \|\theta^*\|_1} . \tag{42}$$

This bound is only useful for $\|\theta^*\|_1 < 1$. Otherwise, we want to apply Lemma B.3. However the values of $\varepsilon'$ and $d_0 = \|\theta^*\|_0$ are unknown. We approximate them with $\tilde{\varepsilon}$ and $\tilde{d}_0$ on exponential grids, which we define now:

$$\mathcal{G}_{\varepsilon'} = \left\{2^{-k}, \quad k = 0, \ldots, i\right\} \quad \text{and} \quad \mathcal{G}_{d_0} = \left\{1, 2, \ldots 2^{-\lfloor \ln d \rfloor}, d\right\} .$$

We define for all $\tilde{\varepsilon} \in \mathcal{G}_{\varepsilon'}$ and $\tilde{d}_0 \in \mathcal{G}_{d_0}$ the dilated soft-threshold

$$\tilde{\theta}(\tilde{\varepsilon}, \tilde{d}_0) := S_{\tilde{\varepsilon}}(\bar{\theta}^{(i)})\left(1 + \frac{2\tilde{d}_0\tilde{\varepsilon}}{\|S_{\tilde{\varepsilon}}(\bar{\theta}^{(i)})\|_1}\right) \wedge \frac{1}{\|S_{\tilde{\varepsilon}}(\bar{\theta}^{(i)})\|_1} , \tag{43}$$

with the convention $\frac{0}{0} = 0$, recalling the definition of the soft-threshold operator $S_\varepsilon(x)_i = \text{sign}(x_i)(|x_i| - \varepsilon)_+$ for all $1 \leqslant i \leqslant d$. Because $\varepsilon' \geqslant 2^{-i}$ (using $\varepsilon \geqslant \sqrt{a} \geqslant 4$, $G \geqslant 1$ and $\tau \leqslant 1/2$ and $\mu \geq 1$) and $\left\|\bar{\theta}^{(i)} - \theta^*\right\|_\infty \leqslant 1$, there exists $\tilde{\varepsilon} \in \mathcal{G}_{\varepsilon'}$ such that $\tilde{\varepsilon} \leqslant 2\varepsilon'$ and $\left\|\bar{\theta}^{(i)} - \theta^*\right\|_\infty \leqslant \tilde{\varepsilon}$. Furthermore, there exists also $\tilde{d}_0 \in \mathcal{G}_{d_0}$ such that $d_0 \leqslant \tilde{d}_0 \leqslant 2d_0$. We can thus apply Lemma B.3, which yields

$$D(\theta^*, \tilde{\theta}(\tilde{\varepsilon}, \tilde{d}_0)) \leqslant \frac{2\tilde{d}_0\tilde{\varepsilon}}{\|\theta^*\|_1} \leqslant \frac{8d_0\varepsilon'}{\|\theta^*\|_1} . \tag{44}$$

We define the new approximation grid

$$\Theta^{(i+1)} := \left\{\tilde{\theta}(\tilde{\varepsilon}, \tilde{d}_0), \tilde{\varepsilon} \in \mathcal{G}_{\varepsilon'}, \tilde{d}_0 \in \mathcal{G}_{d_0}\right\} \cup \left\{[\bar{\theta}^{(i)}]_{\tilde{d}_0}, \quad \tilde{d}_0 = 1, \ldots, d\right\} , \tag{45}$$

where $\tilde{\theta}(\tilde{\varepsilon}, \tilde{d}_0)$ is defined in Equation (43) and $[\cdot]_k$ are hard-truncations to $k$ coordinates. We get from Inequality (42) and (44) that

$$
\begin{aligned}
D\big(\theta^*, \Theta^{(i+1)}\big) &\leqslant \min\left\{\frac{\sqrt{8d_0}}{1 - \|\theta^*\|_1}, \frac{8d_0}{\|\theta^*\|_1}\right\} \varepsilon' \\
&\overset{(41)}{=} (8\sqrt{a}G\varepsilon)^{\frac{\beta}{2}} \mu^{-\frac{1}{2}} \min\left\{\frac{\sqrt{8d_0}}{1 - \|\theta^*\|_1}, \frac{8d_0}{\|\theta^*\|_1}\right\} 2^{-(\frac{1}{2}+\tau)\frac{\beta}{2}i}.
\end{aligned}
$$

To conclude the induction, it suffices to show that this is smaller then $\varepsilon 2^{-\tau(i+1)}$. Our choices of $\varepsilon$ and $\tau$ defined in (36) was done in that purpose, so that the induction is completed.

*Conclusion.* Substituting the values of $\varepsilon$ and $\tau$ into Inequality (40) and using the choice $i = \ln_2 T$ (which upper-bound the number of sessions after $T$ times steps) concludes the proof:

$$
\begin{aligned}
\mathbb{E}\big[\ell_t(\bar{\theta}^{(i)}) - \ell_t(\theta^*)\big] &\overset{\text{Jensen}}{\leqslant} \frac{R_T^{(i)}}{2^i} \\
&\overset{(40)}{\leqslant} 8\sqrt{a}G\varepsilon 2^{-(\frac{1}{2}+\tau)i} \\
&\overset{(36)}{\leqslant} \max_{\theta^* \in \Theta^*}\left(\frac{128a}{T}\max\left\{\frac{1}{\alpha}, \frac{4G^2\|\theta^*\|_0}{\mu}\min\left\{\frac{1}{(1 - \|\theta^*\|_1)^2}, \frac{8\|\theta^*\|_0}{\|\theta^*\|_1^2}\right\}\right\}\right)^{\frac{1}{2-\beta}},
\end{aligned}
$$

where we recall that $a := 16(1 + \ln(K_i) + \ln\ln(ET^2)) + 4x$, where $K_i := \text{Card}(\Theta^{(i)}) + 2d \leqslant (1 + \ln_2 d)(1 + \ln_2 T) + d$. Summing over $i = 1, \ldots, \ln_2(T)$, we get the upper-bound for the cumulative risk.