[Reviews · NeurIPS 2018]

Reviewer 1



This paper considers online learning with a family of losses defined on the L1 ball with some parametrized curvature. The algorithm automatically adapts to this curvature information. In the case of strongly-convex losses, a variant of the algorithm can adapt to the sparsity (L0 norm, _not_ the L1 norm) of the competitor. The algorithm itself is a somewhat technical modification of an experts algorithm that mainains a set of experts corresponding both to (point, learning rate) pairs, where the points come from some specified subset of the L1 ball. Quality: This paper seems to have nice results. I liked that the algorithm can adapt to the curvature without knowing the curvature parameters, and I was also impressed by the second algorithm's ability to adapt to the L_0 norm of the competitor. I really liked the inductive technique used in the proof of Theorem 3.3. The results of Theorem 3.2 are a little hard to interpret, due to the unusual distance term. It seems clear there is a runtime/regret bound tradeoff at work here, but is there some example for a regret bound that is better than 1/sqrt{T}? It seems that (at the cost of exponenial runtime, as in Theorem 3.1), this should be achieveable. Is there anything better one can do? Clarity: I found this paper a bit difficult to follow at times. In particular, the intuition for the proofs is not presented well, which I found very distressing since the statements seem relatively powerful and the corresponding proofs seem very technical. In the sketch of proof for theorem 3.2, on line 221, it seems to be implied that regrets is linear in the comparison point, which confused me since the loss functions have curvature. I believe this was actually a statement about linearized regret, but this was not clear without some poking in the appendix. In the proof of theorem 3.3, the value "a" is initially defined to have a term \log(3d), but later (line 594), this seems to become \log(2d). In line 89, it is stated that under the Lojasiewicz condition the algorithm obtains low regret for comparitors with l1 norm bounded by c<1. However, the corresponding theorem (which I believe is Theorem 3.4) I believe replaces c with 1-\gamma. It seems best to use the same notation. Equation (7) in the appendix is stated to be a restatement of Theorem 3.2 of Wittenberger(2017), but it is not obvious to me that it is the same as it seems to be missing a log(E_\pi_0[1/\eta]/E_{\pi[1/\eta]}). No doubt I'm missing something here, but the proofs are technical enough that I'd appreciate even small gaps being filled. The use of NP to describe an algorithm in line 175 was also a bit confusing to me. The algorithm proposed is exponential time. But to be precise, algorithms are not in NP - problems are. And this problem seems not to be since it is solved in polynomial time in the next theorem. Maybe there is some other meaning of NP, but in general the meaning is clear if you just say exponential time. Also, there are some shenanigans with the typesetting. The paper is over the length limit, and it appears to have smaller text and bigger margins than it should. Finally, there are few minor grammatical errors 84: "it exists" -> "there exists" 159: "it is worth to point out" -> "it is useful to point out" 175: sentence starting with "In Section 3.1..." seems to switch between plural and singular tenses several times. Originality: I believe these results are original. Significance: The adaptivity of the algorithm to the curvature and L0 sparsity is interesting. I think the analysis of the algorithm of section 3.3 had some interesting techniques. Overall I think there is reasonable hope the techniques may be used to design future algorithms.

Reviewer 2



Summary The setting of this paper is the Online Convex Optimization setting, in which a forecaster is to predict a vector from a convex set and the goal is to minimize the average risk w.r.t. some oracle parameters. This paper develops O(dT) runtime algorithms that achieve an average risk with fast and slow rates (depending on properties of the loss function) that scale with the sparsity of the oracle. The approach is to discretize the L_1 ball and learn the optimal parameters in the discretization of the L_1 ball using squint with a discrete prior / boa. This yields a quantile bound on the average risk for the discretized set. Then, using a new pseudo-metric called "averaging accelerability" the risk for the L_1 ball is split into two parts: the risk for the discretized set plus the averaging accelerability times O(log(cardinality discretized set)/sqrt{T})). Now, depending on the discretization and the loss functions this yield rates ranging from O(1/\sqr{T}) to O(1/T). The authors provide two ways of adaptively learning the discretization depending on whether the data are i.i.d. or adversarial, which leads to two new algorithms, SABOA and BOA+. The approach is to restart squint/boa every time the discretized set is updated. The discretized set is updated in such a manner that it approaches the set of minimizers, which leads to a small averaging accelerability. Strengths and weaknesses This paper is well written and clearly identifies the core technical ideas required to arrive at the main results. The build-up of the ideas going from squint/boa on a discretized set to averaging accelerability to learning the discretized set gradually guides the reader through the ideas used to derive the results in this paper. However, the proofs of the main results can be quite laborious. I quite liked the new averaging accelerability pseudo-metric and how it ties in with the analysis. It appears to be a useful new notion to measure the distance between the discretized set and the oracle for sparse oracles. The fact that SABOA and BOA+ achieve fast rates for sparse oracles while maintaining a O(dT) runtime is interesting as thus far only non-efficient procedures were able to achieve this. Minor comments - footnote 3 has disappeared. - there dot at the end of algorithm 1 falls on next line - proposition 1 appears to be missing the probability statement - line 150: evaluated thanks to a loss function -> evaluated by a loss function - line 173: I can not make sense of "one has to add in the grid of experts some points of the l_1-ball to the 2d corners - line 267: "Bernstein condition: it exists \alpha' > 0" it exists should be changed so that this makes sense - line 488: V_k/a -> V_k/a_k - line 526: I believe one still needs to apply Jensen's inequality to conclude the proof

Reviewer 3



# Update after reading the author response Thank you for the explanations. I have improved my score. I hope the following points help the authors see why I am still not enthusiastic about acceptance. 1- I still think the comparison to previous work is inadequate. For example, the SAEW algorithm of [12] uses several similar techniques (like BOA plus sequential restarting), and achieves a bound (in a less general setting) that scales with $d_0$ rather than $d_0^2$. (it was only after discussion with other reviewers that I realized your sentence in the Conclusion - "These rates are deteriorated compared with the optimal one that require restrictive assumption." - is perhaps referring to this difference). How do your techniques differ? How do your running times differ? I had suggested a table summarizing the previous work and where your work stands w.r.t. them in terms of assumptions, rates, and complexity, but from the author response, it feels like the authors are mainly looking into only moving the contributions earlier in the intro. 2- I still believe it is hard for the reader to see the general view of your paper. All the relevant points are there, but the context is missing or is lost between seemingly smaller improvements here and there (e.g. relaxing sc to exp-concave or Lojasiewic). It would have been much nicer if the intro started by saying why we need quantile bounds (instead of getting to it only in L138), what others had done (e.g. [18,31,22]), and what specifically was missing in previous work (like the efficient adversarial alg). The reader of course knows that you want to "obtain lower bounds than any existing online algorithm in sparse setting", but they first need to see an overview of what existed and what was missing to be able to understand in which direction you will be extending the previous work. A good example of such an introduction is maybe that of [18], among many others. Once this is established, then the contribution section could highlight your most important ideas (more like expanding on the list you have at the end of your introduction section, as opposed to attending to the details of the Lojasiewic's assumption). In this way, it is also easier for the reader to understand (at a high level) your improvements over previous work, because you have already talked about their limitations. 3- I think the 8-page limit should not result in the paper not being self-contained. Relevant results and extra comments could be recalled / put in the appendix, and short explanations could be given to connect things. I hope that the authors consider these points in the final version if the paper is accepted, or in the subsequent submissions if it is not. # Summary The paper seems to provide new projection-free algorithms for online learning over the l1-ball with guarantees that depend on the sparsity of the optimal parameter / competitor. This is what *seems* to be the main contributions of the paper: 1- An algorithm (Alg-1) for the expert-advice setting that enjoys (per Thm 2.1) a quantile bound, which scales with O(1/T) when the losses are exp-concave, improving / extending previous work [18, 31, 22]. 2- A projection-free algorithm (BOA+) for online strongly-convex optimization over the l1 ball, by restarting Alg-1 on grids that include new leaders (in the Follow-The-Leader sense) from the previous rounds. 3- Another algorithm (SABOA) for stochastic strongly-convex optimization over the l1 ball, similar to BOA+ but with a different grid construction. 4- O(1/T) rates for BOA+ and SABOA that depend on the number of non-zero elements of the competitor / optimal parameter. However, the paper is generally hard to read, in the sense that it is not self-contained (at all), is not coherently written, comparison to previous work is scattered throughout the paper, and and the overall message is hidden by the details. In particular, the high-level view of the paper, the extent of technical novelty, the exact relation to previous work, and if / why the bounds are optimal in each of the problem parameters (i.e., sparsity of the competitor, Lipschitz and exp-concavity constants, etc) is not entirely clear. As a result of the inaccessibility of the paper, I do not recommend publishing at NIPS. ## Questions for the authors Please consider answering Q1-Q5 below in the author response. (Please consider answers that shed light on the general view and significance of your work). # Quality and Significance From a technical point of view, the contributions seem to be interesting. However, the quality of the presentation suffers, to the point of making the results inaccessible, except to the reader with direct experience working with the underlying algorithms / ideas. ## Algorithm 1 Alg-1 is the Squint algorithm of [18] with a geometric prior on learning rates. The result seems to improve over [31, Section 4] by requiring only exp-concavity of the losses (in expectation) as opposed to strong convexity. In addition, the KL-div is not simplified to log(K) - as [31] does in their Thm 4.3 for example, so quantile bounds are obtained. It extends [18] (and the relevant prior work that obtain O(1/sqrt(T)) quantile bounds, not cited here) by providing the O(1/T) rate for exp-concave losses. It seems to improve over [22] by being a completely online algorithm and removing a gap term in the bound. **Q1**: Where similar priors used for Squint before? Compared to [31], are there any other technical novelties I have not mentioned above? What is the challenge of replacing strong convexity with weak exp-concavity (wouldn’t very similar techniques as in [31, Sec 4] directly apply here?) **Q2**: Why is the weakening of exp-concavity important? Can you provide concrete examples of useful losses where the weak exp-concavity holds, but exp-concavity doesn’t (i.e., what are the “regularity conditions”?) ## BOA+ and SABOA BOA+ and SABOA extend Alg-1 to the l1 ball (Alg-1 is for a finite set of experts) by restarting it, for exponentially increasing number of time steps, on descritized grids that gradually include improved approximations of the optimal parameter. The algorithms are interesting as they are projection-free, yet obtain the O(1/T) rates of convergence under strongly-convex (or Lojasiewicz) losses. Again, the lack of coherence and proper comparison with previous work prevents complete appreciation of the results. It seems that the main benefit over previous work are that the algorithms are projection-free, and the Lojasiewicz condition is used, generalizing strong-convexity to a class of non-convex functions. Also, the same bounds apparently apply to general convex functions as well (reducing to O(1/sqrt(T)) in that case). **Q3**: How do your results (Sec 3) benefit from the exp-concavity condition, as you still need to assume strong-convexity, or it’s non-convex counter-part (Lojasiewicz’ condition)? Would they also benefit from the weakening of exp-concavity in A2? **Q4**: Regarding [26] and [12], isn’t competing with the minimizer over R^d harder than with the minimizer of B_1, as the former always has a lower loss than the latter, and thus the bounds will be stronger? **Q5**: How do the computational complexity of BOA+ / SABOA compare with previous work? Are [13] and [32] the only relevant work here? How do the bounds compare? Is the dependence on each of the problem parameters (|| Theta ||_0, G, alpha, beta, mu, etc) optimal? What / where are the relevant lower bounds? # Clarity I found the paper hard to read, even for someone with experience in online learning, the expert advice framework, bandits, etc. The reader needs deep familiarity with specific previous work (Squint, BOA, etc). While the relevant sections are there (contributions, previous work, etc.), one has to read the paper multiple times, as well as the related work, to find out what the actual contributions are, why they are important, how they compare with previous work, etc. A few examples are provided below: 1- I recommend consulting a professional editing service to improve general readability. There are several sentences that are not informative, or are confusing. It is not clear what some pronouns refer to. For example: “These results generalize previous work [..]. **They** are obtained under [..] Lojasiewicz condition.” Which results? All of them? But this seemed to relate to convex / exp-concave functions. One has to read until the end to find out which results is given under what condition. 2- The introduction should discuss what is the goal of the paper, what are the high-level results, and why they matter and were not obtained before. Instead, it gets to details such as using Dirac masses to get the regret, and goes to “Previous work”. What do you want to do? Why? 3- The previous work section seems incomplete, uninformative, and vague. What is “a subroutine centered around the current estimate”? Why should we know this? We still don’t know what the paper is trying to do… Shouldn’t Lines 41,42 actually just go to the “contributions” section? 4- Similar vague sentences and paragraphs appear throughout the paper. The paper is not self-contained (e.g., “extended to a unique approximately sparse Theta* [ ..] see [1;3]”). Hand-wavy explanations and phrases are used where appropriate mathematical expressions should appear (e.g., “set of hard-truncated and dilated soft-thresholded versions of [..] as in (43)”). On the other hand, explanation is absent where it would be useful. For example, why should one consider SABOA instead of BOA+ in the stochastic setting? Why can we use previous iterates instead of the leaders here? # Originality The comparison with related work is unclear. The reader needs to go to the related work and find the results and compare them with the current results (even, e.g., the respective Theorem numbers are not given). I recommend providing a table comparing previous work (e.g., as in [22]), showing whether each result concerned regret or cumulative risk [as in 22], strongly-convex, exp-concave, or weak exp-concave losses, was it an h.p. or expectation bound, etc.